# Generating Highly Designable Proteins with Geometric Algebra Flow Matching

**Simon Wagner**[* 1,2]    **Leif Seute**[* 1,2,3]    **Vsevolod Viliuga**[1,3,4]    **Nicolas Wolf**[1,2,3]
**Frauke Gräter**[1,2,3]    **Jan Stühmer**[1,5]

[1]Heidelberg Institute for Theoretical Studies, Heidelberg, Germany
[2]IWR, Heidelberg University, Heidelberg, Germany
[3]Max Planck Institute for Polymer Research, Mainz, Germany
[4]SciLifeLab and DBB at Stockholm University, Stockholm, Sweden
[5]IAR, Karlsruhe Institute of Technology, Karlsruhe, Germany

## Abstract

We introduce a generative model for protein backbone design utilizing geometric products and higher order message passing. In particular, we propose Clifford Frame Attention (CFA), an extension of the invariant point attention (IPA) architecture from AlphaFold2, in which the backbone residue frames and geometric features are represented in the projective geometric algebra. This enables to construct geometrically expressive messages between residues, including higher order terms, using the bilinear operations of the algebra. We evaluate our architecture by incorporating it into the framework of FrameFlow, a state-of-the-art flow matching model for protein backbone generation. The proposed model achieves high designability, diversity and novelty, while also sampling protein backbones that follow the statistical distribution of secondary structure elements found in naturally occurring proteins, a property so far only insufficiently achieved by many state-of-the-art generative models.

## 1 Introduction

Recent years have shown tremendous progress in applying deep learning to computational chemistry, where applications of learning-based approaches have enabled unprecedented progress across a broad range of problems, such as molecular property prediction [55, 28, 50, 11], protein-ligand docking [22], protein structure prediction [32, 3, 38], and *de novo* protein design [63, 68, 10, 67, 37]. In case of protein design, state-of-the-art methods typically represent the structure of a protein of $N$ residues as an element of $SE(3)^N$, *i.e.* as a collection of $N$ frames, each of which describes the position and orientation of an individual protein residue. Among the most successful methods are those based on diffusion models [53, 54, 63, 68, 37, 1] and flow matching [39, 19, 56, 67], which make use of architectures that incorporate the invariant point attention (IPA) of AlphaFold2 [32]. By modeling a protein through the frames of its backbone, the task of protein structure generation is reformulated as to model the distribution of a set of frames, an inherently *geometric* problem.

Advances in geometric deep learning have lead to architectures that are equivariant towards rotations and translations [14, 21, 15], which can be regarded as a geometric inductive bias that enhances performance and data efficiency. Most generative models for protein design achieve equivariance by using geometric features that are expressed in the canonical local coordinate frames representing the protein backbone. The coordinates of these features are thus invariant with respect to global rotations and translations, which allows to apply general layers and non-linearities.

---

[*]Equal contribution

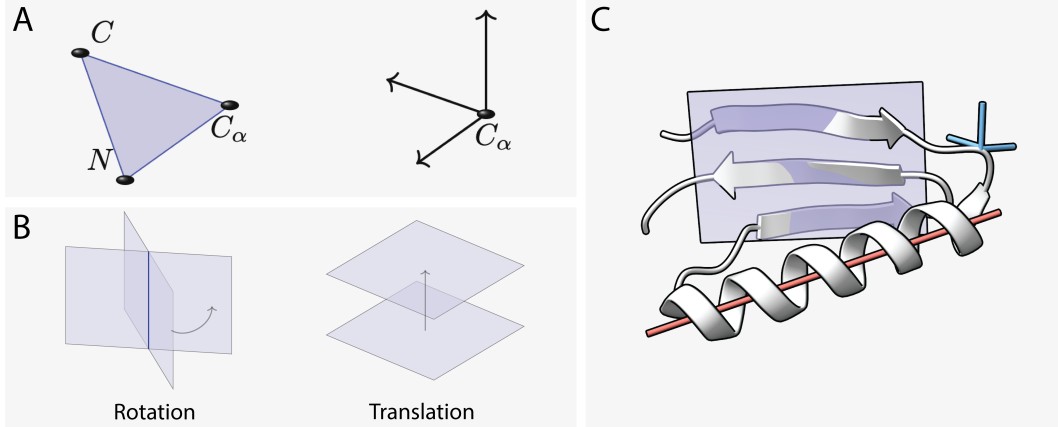

Figure 1: (**A**) Protein backbone residue with three backbone atoms represented by a coordinate frame. (**B**) In PGA, a frame can be represented via the geometric product of four planes. Two of the planes parameterize the frame's rotation around their line of intersection, while the other two encode the frame's translation along the separation vector between them. (**C**) An exemplary protein backbone structure containing an $\alpha$-helix and a $\beta$-sheet. Lines (red), planes (violet) and Euclidean frames (blue) can all be embedded as elements of PGA, facilitating a geometric inductive bias for learning representations of the abstract geometry of the protein.

Another widely used approach to construct equivariant architectures is to embed internal features in symmetry group representations and restrict neural network operations to equivariant functions [55, 28, 50, 11, 7, 6, 36, 52]. Explicit equivariance in these models has usually been limited to the orthogonal group $O(3)$, which contains rotations and reflections, while most models are instead invariant towards translations. Only recently, Brehmer et al. [13] proposed an architecture based on the framework of Ruhe et al. [48] that enables explicit equivariance towards both translations and rotations by utilizing the projective geometric algebra (PGA) [29, 25].

Inspired by their work, we demonstrate that apart from its use as E(3)-equivariant formalism, PGA provides a powerful framework for representing the geometry of protein backbones when incorporated into the local frame formulation of protein design architectures. While we utilize PGA to explicitly represent the frames of the protein backbone as elements of the algebra, PGA additionally provides a strong inductive bias as its elements can represent abstract geometric objects like points, lines and planes, which are well suited to capture the geometry of secondary structure elements, such as $\alpha$-helices and $\beta$-sheets (Figure 1). Moreover, bilinear operations of the algebra enable to compute many geometric relations between those objects, such as distances, angles, projections and incidences.

**Main contributions:**   We introduce *Clifford frame attention* (CFA), an extension of the invariant point attention (IPA) architecture of AlphaFold2 [32] by expressing geometric features as elements of the projective geometric algebra and using its bilinear operations to construct geometrically expressive, higher order messages. We incorporate CFA into an existing flow matching framework for protein backbone generation, FrameFlow [67], and demonstrate in our experiments that the proposed method achieves state-of-the-art performance in the combination of designability, diversity and novelty of generated protein samples. While, especially for small proteins, other models with high designability often over-represent $\alpha$-helices, the proposed method captures the broad distribution of secondary structures of naturally occurring proteins, which we believe is crucial for designing proteins with vast functionalities. Notably, since IPA is widely used throughout the field, CFA may be readily included in many other protein-related machine learning models.

## 1.1   Related Work

**Geometric (Clifford) algebra in neural networks**   Neural networks that use the Clifford algebra were first proposed by Pearson and Bisset [46], an extension of the multi layer perceptron (MLP) by Clifford algebras, which was later studied further by Buchholz and Sommer [16]. More recently, Ruhe et al. [49] propose Geometric Clifford Algebra Networks, using geometric (Clifford) algebras based on their Clifford Neural Layers [12], and extend this framework in [48] to E(3)-equivariant

representations. Brehmer et al. [13] propose to use projective geometric algebra, which enables SE(3)-equivariant feature representations, and the representation of frames as elements of the algebra.

**Generative models for protein design** Watson et al. [63] propose RFdifffusion, a generative model for protein backbone design that utilizes the pre-trained protein structure prediction network RoseTTAFold [2]. Yim et al. [68] propose FrameDiff, which defines a diffusion model over a set of frames, $SE(3)^N$, and extend this method within the flow matching framework [67]. Bose et al. [10] propose FoldFlow and its variants, diffusion and flow matching models over $SE(3)^N$. Lin and AlQuraishi [37] propose an equivariant encoder and decoder architecture. Wu et al. [65] train a transformer model to predict angles between adjacent residues. Mao et al. [42] propose vector field networks, which is also an extension of IPA, and employ them in FrameDiff. The main difference to our approach is that VFN uses virtual atoms as geometric features and *vector field operators* to model interactions, whereas CFA use multivectors and the geometric bilinears of PGA respectively.

## 2 Background

This section provides an introduction to the mathematical frameworks that we refer to in this paper, flow matching and Geometric algebra, and a brief introduction to protein design.

### 2.1 Geometric Algebra

A Clifford algebra over a real vector space, typically referred to as geometric algebra, is a powerful mathematical framework for describing geometric objects including points, lines, planes and operations on these objects in an algebraically concise way [31, 26].

In more technical terms, given a vector space $V$ and a quadratic form $q : V \to \mathbb{F}$ from the vector space to the underlying field $\mathbb{F}$, we can construct a geometric algebra as the unitary, associative, non-commutative algebra with the property $\mathbf{v}^2 = q(\mathbf{v})$ for every $\mathbf{v} \in V$ [49]. $\mathbf{v}^2 = \mathbf{v}\mathbf{v}$ denotes the *geometric product*, the bilinear operation of the algebra, of the vector $\mathbf{v}$ with itself. In the geometric context, $q$ may be thought of as the metric of the vector space, meaning that for a vector $\mathbf{v}$, $q(\mathbf{v})$ is its squared norm. Elements of the algebra are called *multivectors*. They can be constructed by forming geometric products between basis vectors of $V$ and linearly combining them.

In this paper we are mainly interested in the projective geometric algebra (PGA) [29, 25], which we denote as $\mathbb{G}_{3,0,1}$. It is the geometric algebra over $V = \mathbb{R}^4$ with vector basis

$$\mathbf{e_0}, \mathbf{e_1}, \mathbf{e_2}, \mathbf{e_3} \text{ and } q(\mathbf{e_1}) = q(\mathbf{e_2}) = q(\mathbf{e_3}) = 1, \ q(\mathbf{e_0}) = 0. \tag{1}$$

The full algebra has 16 basis elements, which can be grouped in grades according to the number of vector basis elements they are constructed from. Following Gunn [29], we can interpret elements of different grades as geometric objects such as planes, lines and points in $3D$ space, as listed in Table A.1 and visualized in Figure A.3. Working in a four dimensional space has the advantage that we are able to represent lines and planes that do not necessarily include the origin, which is crucial for the description of translations as shown below.

Central to the proposed method is the fact that PGA allows the representation of elements of the Euclidean group E(3) as elements of the algebra. Given a plane $\mathbf{p} \in \mathbb{G}_{3,0,1}$, an arbitrary geometric object $\mathbf{X} \in \mathbb{G}_{3,0,1}$ can be reflected across the plane via the sandwich product,

$$\mathbf{X}' = \mathbf{p}\hat{\mathbf{X}}\mathbf{p}, \tag{2}$$

where $\hat{\mathbf{X}}$ is the grade involution that flips the sign of elements with odd grade. A vector in PGA may thus be interpreted both as a plane and as a reflection operator. One can extend this idea using the Cartan-Dieudonné theorem, which states that any E(3) transformation can be represented by repeated reflections. Two consecutive reflections through intersecting planes result in a rotation around the line of intersection and two reflections through two parallel planes correspond to a translation along the separation vector of the planes. The associated elements of PGA are obtained by taking the geometric product of the respective planes. We thus represent each residue frame of a protein by a multivector, a so called *motor*, corresponding to a rotation followed by a translation as shown in Figure 1. Similar to eq. 2, a motor $\mathbf{M} \in \mathbb{G}_{3,0,1}$ can be applied to an arbitrary element $\mathbf{X} \in \mathbb{G}_{3,0,1}$ according to

$$\mathbf{X}' = \mathbf{M}\mathbf{X}\mathbf{M}^{-1}. \tag{3}$$

We provide a more detailed introduction to geometric algebra in Appendix A.1.1

## 2.2 Flow Matching

Flow matching [39], as a generalization of diffusion models [53, 54], offers a framework for learning continuous normalizing flows (CNFs) [20], $\phi\colon [0,1] \times \Omega \to \Omega$ that transform a general prior distribution $p_0$, defined on the domain $\Omega$, to a target distribution $p_1$ by evaluating the probability path

$$p_t = [\phi_t]_* p_0 \tag{4}$$

at $t = 1$, where $*$ denotes the pushforward. The flow $\phi_t$ can be expressed in terms of a vector field $v_t$ by solving the ordinary differential equation (ODE)

$$\frac{\mathrm{d}}{\mathrm{d}t}\phi_t(x) = v_t(\phi_t(x)), \quad \phi_0(x) = x\,. \tag{5}$$

If the vector field $v_t$ is known, one can generate samples from $p_1$ by sampling from the prior and integrating the flow ODE. Lipman et al. [39] showed that $v_t$ can be learned by regressing on a vector field $u_t(x|x_1)$ that is conditioned on a data sample $x_1 \sim p_1$, i.e. by minimizing the loss function

$$\mathcal{L} = \mathbb{E}_{t\sim\mathcal{U}(0,1), x_1\sim p_1, x_t\sim p_t(\cdot|x_1)}\left[\|v_t(x_t) - u_t(x_t|x_1)\|^2\right]\,. \tag{6}$$

Here, $p_t(x|x_1)$ is the conditional probability path induced by $u_t(x_t|x_1)$, which is commonly chosen as linear interpolation between the prior sample $x_0$ and the target $x_1$ by setting the conditional flow to

$$\psi_t(x_0|x_1) \equiv (1-t)x_0 + tx_1\,, \tag{7}$$

from which $u_t(x|x_1)$ can be obtained by forming the time derivative. As shown by [39, 19], sampling from $p_t(x_t|x_1)$ in eq. 6 can then be realized by transforming a sample $x_0$ from the prior $p_0$ to $x_t \equiv \psi_t(x_0|x_1)$. This formulation in terms of conditional distributions allows to learn dynamic optimal transport (OT) plans from prior to target using minibatch OT as shown by Tong et al. [56].

The framework of CNFs and flow matching can be generalized for sampling from distributions on Riemannian manifolds $\mathcal{M}$, such as $\mathrm{SE}(3)^N$, as described by Chen and Lipman [19]. Then, the flow $\phi\colon [0,1] \times \mathcal{M} \to \mathcal{M}$ is a diffeomorphism, and a vector field is learned as a smooth map $u_t\colon [0,1] \times \mathcal{M} \to \mathcal{TM}$ to the tangent space $\mathcal{TM}$. A natural choice for the map $\psi_t(x_0|x_1)$ on a Riemannian manifold is the connecting path with minimal length, the geodesic.

## 2.3 Flow Matching for Protein Structures

For sampling from the distribution of protein backbones, we represent each protein residue by a frame $T = (r, x) \in \mathrm{SE}(3)$ that corresponds to a translation $x \in \mathbb{R}^3$ and rotation $r \in \mathrm{SO}(3)$, thus the domain for the flow matching process is $\mathcal{M} \equiv \mathrm{SE}(3)^N$, as described in FrameDiff [68]. The frame for a given residue is defined through the backbone atoms $[N, C_\alpha, C]$, where the translation $x$ is chosen as the displacement of the $C_\alpha$ atom relative to the origin, while the rotation $r$ is constructed using a Gram-Schmidt process [32] on the vectors $[C - C_\alpha, N - C_\alpha]$.

Similar to [67], we define the conditional flow $\psi_t(T_0|T_1)$ as the geodesic between $T_0$ and $T_1$,

$$\psi_t(T_0|T_1) = \exp_{T_0}(t \log_{T_0}(T_1))\,. \tag{8}$$

We calculate the time derivative of the flow as described in FrameFlow [67] and regress on it as in eq. 6. As prior $p_0$, we choose a $3N$-dimensional normal distribution with unit variance $\mathcal{N}(0, I)$ for translations and, for rotations, adopt the heuristic trick of using $\mathcal{IG}SO(3)$ during training and $\mathcal{U}(SO(3))$ for inference from Yim et al. [67]. Additionally, we use minibatch optimal transport [56] with the equivariant cost function from [34, 10].

## 2.4 Designability of Proteins and Importance of their Structural Composition

Until recently, computational tools for protein design have relied on physics-based energy functions [51, 35, 64], restricting protein design campaigns to pre-defined topologies, geometries, and secondary structure elements. In contrast, deep generative models learn the probability distribution of protein structures from the training set, enabling less constrained sampling of the structure space.

Ideally, the main goal of any generative model should be the creation of structurally diverse and novel proteins, thereby maximizing the access to unseen structures and possibly functions. Since

the functionality of proteins is grounded in their structure, it is crucial to extensively cover the broad range of *secondary structure* elements and topologies found in natural proteins. Typically, the properties of generated proteins are assessed by the scores of *diversity* and *novelty*, indicating how similar generated backbones are to each other, and to known native proteins, respectively. While explicit analyses of secondary structure elements of generated proteins are often overlooked, [37, 65] suggest that diffusion models commonly generate redundant protein structures composed mostly of $\alpha$-helices, which raises the question of how much functionality can be hypothetically encoded in these proteins.

A key quality check for generated proteins is their biophysical consistency in the sense that their sequence, indeed, folds into the intended structure [44]. In the case of backbone generation, this is typically evaluated by predicting a sequence from the generated structure with the inverse folding model ProteinMPNN [23]. The obtained sequence is re-folded with ESMFold [38] and if the resulting structure aligns well with the original backbone, the latter is called *designable* [57, 63]. Thus, the designability metric measures consistency between well-established sequence and structure prediction models, which can be seen as a proxy for biophysical validity and practical realizability [47].

## 3    Geometric Algebra Flow Matching for Protein Backbone Generation

We propose a geometric algebra-based neural network architecture to predict the vector fields in a flow matching process on $SE(3)^N$ and call this approach Geometric Algebra Flow Matching (GAFL). The proposed neural network architecture is an extension of the one introduced in FrameDiff [68], where we replace its central component, the invariant point attention (IPA) block from AlphaFold2 [32] by *Clifford Frame Attention* (CFA), which we explain in more detail in the next paragraph.

Input features are the noised frames $T_t$, pairwise spatial distances, positional encodings of absolute and relative sequence positions, and the flow matching time $t$. As FrameDiff, we use self-conditioning. The network relies on a series of six blocks, in which the frames are updated consecutively to predict the denoised backbone structure and from this the respective conditional vector fields. Each block uses CFA to perform message passing between protein residues, in particular processing geometric information as given by geometric node features and the current set of frames. The $SE(3)$ invariant output of CFA is fed through an MLP and a transformer [60] and then used to predict frame updates. For a detailed explanation of the architecture, we refer to Appendix A.2.1.

**Clifford Frame Attention**

The original IPA mechanism uses geometric node features in the form of 3D points for the calculation of attention scores and as queries, keys and values, as described in Appendix A.2.2. The features are expressed in local coordinate frames, which allows the use of arbitrary layers without breaking equivariance. Messages between nodes are constructed as a linear combination of attention values, weighted by attention scores. While other generative models for protein design incorporate the original version of IPA directly [68, 67, 10], we propose to enhance its geometric expressivity by performing the following modifications.

**PGA features:**   We replace the point-valued attention values by multivectors, which can encode points, lines, planes and Euclidean frames as shown in Figure 1. At the same time we decrease the number of channels to retain approximately the same number of parameters.

**Geometric messages:**   Instead of linearly combining geometric features in the message passing step, we construct messages utilizing the geometric bilinear layer that was introduced as part of the Geometric Algebra Transformer in Brehmer et al. [13], effectively calculating geometric product and join, another bilinear operation of PGA (see Definition A.10), between the node features $\mathbf{V}_i$ and $\mathbf{V}_j$ (see Algorithm 4). These two operations are able to compute many geometric relations as detailed in Appendix A.1.1 and [25]. The messages $m_{ij}^{h,p}$ from node $i$ to node $j$ can then be written as

$$m_{ij}^{h,p} = \text{GeometricBilinear}\left(\mathbf{T}_i^{-1}\left(\mathbf{T}_j\mathbf{V}_j^{h,p}\mathbf{T}_j^{-1}\right)\mathbf{T}_i, \ \mathbf{V}_i^{h,p}\right), \tag{9}$$

where $\mathbf{T}_i, \mathbf{T}_j$ are the frame transformations of the corresponding node, which ensure that the bilinear operations are performed in the same reference frame. When aggregating the messages we make use

of the bilinearity of the products to exchange the sum and the bilinear layer to obtain:

$$\sum_j a_{ij}^h m_{ij}^{h,p} = \text{GeometricBilinear}\left(\mathbf{T}_i^{-1}\sum_j a_{ij}^h\left(\mathbf{T}_j \mathbf{V}_j^{h,p}\mathbf{T}_j^{-1}\right)\mathbf{T}_i,\ \mathbf{V}_i^{h,p}\right),\qquad(10)$$

The operation is thus performed on a node level and scales linearly with the number of nodes.

**Higher order message passing:**  We also incorporate higher order messages into the proposed architecture, that is, messages that depend on more than two nodes and are thus capable of describing relationships beyond pairwise interaction. As in [6], we use bilinearity of geometric product and join to construct higher order messages by multiplying aggregated two-body messages $m$ and $m'$, e.g.

$$\left(\sum_j m_{ij}\right)\left(\sum_k m'_{ik}\right) = \sum_{jk} m_{ij}m'_{ik} \equiv \sum_{jk} m_{ijk}^{(3)} \qquad(11)$$

can be seen as sum of three-body messages $m_{ijk}^{(3)}$. In practice, we use a learnable linear combination of versions of eq. 11 projected onto different multivector grades using the geometric product and the join, as described in algorithm 5.

**Frames as features:**  Since we embed both residue frames and geometric node features in $\mathbb{G}_{3,0,1}$, it is a natural idea to combine them on a node level such that they can interact via the geometric bilinears during message passing. To this end we compute relative frame transformations for all pairs and aggregate them with the attention weights,

$$\mathbf{T}_i^{\text{rel}} \equiv \sum_j a_{ij}\mathbf{T}_i^{-1}\mathbf{T}_j\,. \qquad(12)$$

We concatenate these with the remaining geometric features before the construction of geometric messages and also pass them directly to the backbone update block using a residual connection. The full CFA algorithm is provided in Appendix A.2.3.

Although PGA would in principle allow to construct a fully equivariant architecture ([13]) without using local frames, we decide to keep the local frame formulation of IPA, since it allows to use more general layers and non-linearities. Moreover, the common problem of ambiguous local frame choices that other architectures suffer from [62] is not apparent in our case since backbone residues provide a canonical, geometrically meaningful choice for the local frames. We discuss the equivariance of GAFL in Appendix A.2.4.

## 4 Experiments

We train GAFL[2] on a subset of the Protein Data Bank (PDB) dataset [9] comprised of monomeric protein structures with up to 512 residues and perform extensive ablations on the smaller, curated SCOPe dataset [27, 18] filtered by proteins with length of up to 128 residues (SCOPe-128) as in [67, 37]. A representative selection of designable protein backbones generated by GAFL trained on the PDB dataset is illustrated in Figure 2.

### 4.1 Checkpoint Selection

With the aim of finding a good balance between designability and secondary structure content in mind, we introduce a *checkpointing criterion* that takes secondary structure into account. We first train the model for $N_{\text{train}}$ epochs. Then, from epoch $N_{\text{train}}$ to $N_{\text{train}} + N_{\text{select}}$, we calculate the relative occurrence of $\alpha$-helices and $\beta$-strands $r_\alpha$ and $r_\beta$ of 100 generated proteins after each epoch. We keep the top $k$ checkpoints in terms of secondary structure content deviation, which we define as $d_c \equiv |r_\alpha - r_\alpha^{(\text{ref})}| + |r_\beta - r_\beta^{(\text{ref})}|$, with the training set as reference. Among those $k$ checkpoints, we choose the checkpoint with the highest designability and filter by a threshold $d_c \leq d_{\max}$. The hyperparameters $N_{\text{train}}$, $N_{\text{select}}$, $d_{\max}$ and $k$ for the different training runs are listed in Appendix A.3.8.

---

[2]Source code and trained model weights are available at `https://github.com/hits-mli/gafl`

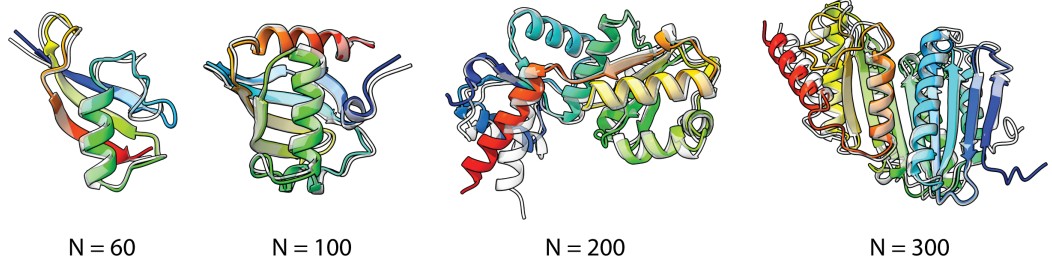

| N = 60 | N = 100 | N = 200 | N = 300 |

Figure 2: Representative examples of designable protein backbones generated with GAFL (white) and the output of the refolding pipeline (colored), comprising ProteinMPNN and ESMFold.

## 4.2 Metrics

To assess the performance of a given model, we follow a well-established pipeline of self-consistency evaluation [57, 63]. For each generated backbone, we design 8 candidate sequences with Protein-MPNN [23], which are subsequently refolded with ESMfold [38], and define scRMSD as the smallest RMSD between our generated backbone and the 8 refolded, aligned candidates. As in [63, 68], we define *designability* as the fraction of generated samples with scRMSD < 2.0 Å. We also report *diversity* and *novelty* of the designable backbones as average TM-similarity [70] scores within the set of generated backbones and with respect to the PDB correspondingly (see Appendix A.3.1 and [67]). To evaluate how well the secondary structure distribution of the training set is captured, we calculate the *average helix and strand content* of all designable backbones using the DSSP algorithm [33].

## 4.3 Baselines

At the task of generating backbones of up to 300 residues, we compare GAFL trained on the PDB to the diffusion models RFdiffusion [63] and FrameDiff [68] and to the flow matching models FoldFlow [10] and FrameFlow [67, 69]. At the time of submitting the paper, FrameFlow had not been trained on the PDB yet, however, we include it in our results as contemporary work. Further, we perform an ablation study for generating smaller backbones, where we compare GAFL models to the originally published FrameFlow [67] model, which was trained on the SCOPe dataset with backbones of up to 128 residues. While all of the baselines above incorporate the original IPA [32] architecture, we also compare GAFL with VFN [42], in which an alternative modification of IPA is proposed. For all models considered, we generate backbones using published model weights and the respective default inference settings (Appendix A.3.2).

## 4.4 Results

We train GAFL for 15 days on two NVIDIA A100-80GB GPUs on the dataset used in FrameDiff, which comprises monomeric structures from the Protein Data Bank (PDB), filtered by a maximum length of 512 and a maximum coil content of 50%, resulting in a total of around 25,000 backbones [68]. As in FoldFlow, we evaluate GAFL and the respective baselines on the task of generating backbones of lengths $\{100, 150, 200, 250, 300\}$. For GAFL, we use 200 inference timesteps (Figure A.9). In Table 1, we report the metrics described in Section 4.2 for each model together with the time needed to generate a single backbone of length 100 on an NVIDIA A100 GPU without batching.

**GAFL has state-of-the-art performance**

We find that GAFL can reliably generate designable, diverse and novel backbones while capturing the statistical distribution of secondary structure elements of natural proteins. In all metrics considered, GAFL outperforms both variants of FoldFlow and is better or as good as FrameDiff. GAFL also outperforms FrameFlow, which was trained on the same dataset, in terms of designability and, at the same time, achieves better secondary structure content. GAFL's designability is only matched by RFdiffusion, which is not directly comparable since it relies on pre-trained model weights from the folding model RoseTTAFold [2] and has around three times more parameters. For diversity, novelty and helix content, GAFL performs better than RFdiffusion. We also observe that GAFL and FrameFlow can generate backbones around three times faster than the other evaluated models.

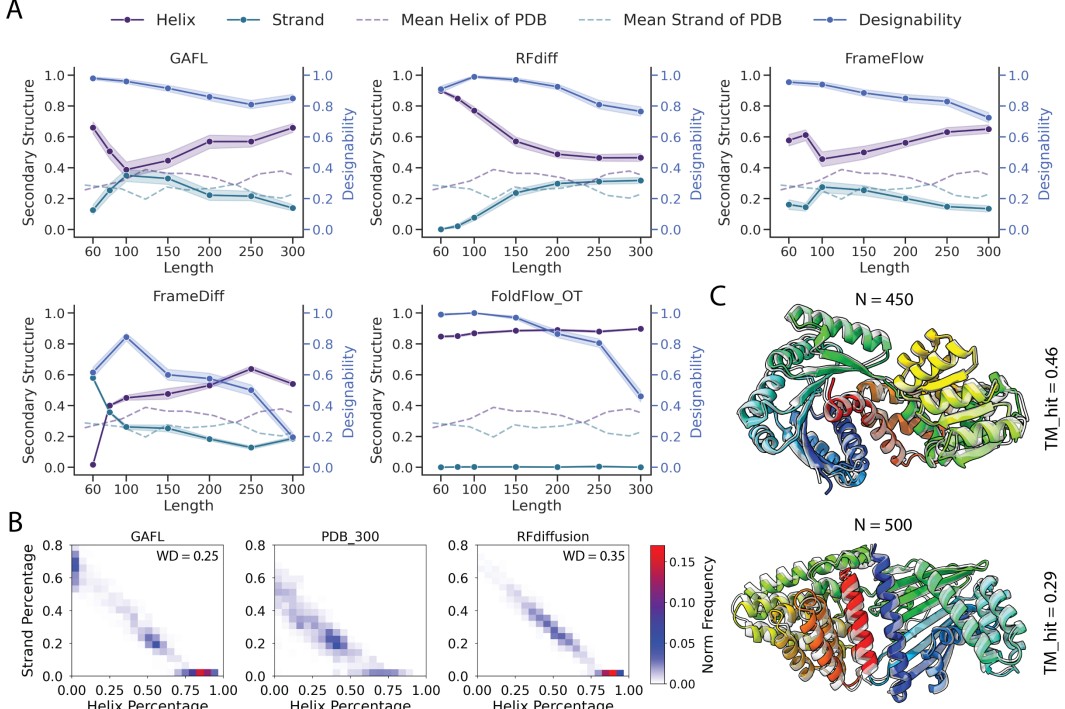

Figure 3: (**A**) Performance of evaluated models in terms of designability and secondary structure content as a function of backbone length. 200 backbones were generated for each model at each length $\in \{60, 80, 100, 150, 200, 250, 300\}$. (**B**) Comparison of the secondary structure distributions of backbones generated by GAFL and RFdiffusion from (A) to the PDB dataset filtered by the respective protein lengths along with the Wasserstein distance (WD) between the distributions. (**C**) Examples of designable backbones generated by GAFL for lengths 450 and 500. We also report TM scores of the backbones to the closest hit in the PDB database computed with FoldSeek.

Table 1: Performance of GAFL and baseline models for the generation of 200 protein backbones for each length in $\{100, 150, 200, 250, 300\}$. We report the metrics from Section 4.2, including standard errors obtained by bootstrapping, and the time needed to generate a backbone of length 100. The best values and values within the respective margin of error are bold.

| Method | Designability (↑) | Diversity (↓) | Novelty (↓) | Helix Content | Strand Content | Time [s] |
|---|---|---|---|---|---|---|
| PDB Dataset (300) | - | - | - | 0.39 (0.00) | 0.23 (0.00) | |
| FrameDiff | 0.54 (0.02) | 0.45 (0.00) | 0.71 (0.00) | **0.53** (0.01) | 0.20 (0.00) | 24.3 |
| FoldFlow-SFM | 0.69 (0.01) | 0.44 (0.00) | 0.77 (0.00) | 0.91 (0.00) | 0.01 (0.00) | 24.3 |
| FoldFlow-OT | 0.82 (0.01) | 0.44 (0.00) | 0.79 (0.00) | 0.88 (0.00) | 0.00 (0.00) | 24.3 |
| FrameFlow | 0.85 (0.01) | **0.35** (0.00) | **0.70** (0.00) | 0.56 (0.01) | 0.20 (0.00) | 6.6 |
| RFdiffusion* | **0.89** (0.01) | 0.37 (0.00) | 0.74 (0.00) | 0.58 (0.02) | **0.24** (0.02) | 21.0 |
| GAFL (ours) | **0.88** (0.01) | 0.36 (0.00) | 0.71 (0.00) | **0.53** (0.01) | **0.25** (0.01) | 8.8 |

*Pretrained weights from folding model trained on dataset larger than PDB.

**GAFL generates proteins with diverse secondary structures at various lengths**

Although protein design campaigns span a wide range of protein sizes [5, 17, 43, 8, 63], the primary goal remains to encode maximum functionality into the smallest protein possible, driven by the growing costs of synthesis as protein size increases. Thus, we further assess designability and secondary structure content of generated backbones as a function of their length (Figure 3A). For all lengths considered, backbones generated by GAFL, FrameFlow and RFdiffusion are highly designable while FrameDiff and FoldFlow struggle with the generation of long proteins. The length dependence of designability and secondary structure content for GAFL and FrameFlow is qualitatively similar; however, GAFL achieves overall better results (Table 1, Figure A.8). Crucially, we find that for generating proteins with less than 150 residues, GAFL is well suited as it is capable of generating

highly designable backbones with a similar amount of $\beta$-strands as naturally occurring proteins, while RFdiffusion over-represents $\alpha$-helices. This is also reflected by the Wasserstein distances between the secondary structure distribution of naturally occurring backbones and those generated by GAFL and RFdiffusion, respectively: Purely $\alpha$-helical proteins are over-represented by RFdiffusion, leading to a higher Wasserstein distance of 0.35 compared to 0.25 for GAFL (Figure 3B).

**GAFL can generate large proteins**

To compare GAFL with VFN [42], we evaluate it at generating five backbones for each length in $\{100, 105, \ldots, 500\}$, some of which are portrayed in Figure 3C. We find that, with a value of 0.74, GAFL outperforms not only VFN (0.44) but also FrameFlow (0.64) and RFdiffusion (0.71) in designability. However, GAFL and FrameFlow over-represent helices for large proteins (Table A.3).

## 4.5  Ablation of CFA

In order to investigate the effect of the proposed architectural changes, we conduct an ablation study on the semi-manually curated SCOPe dataset [27, 18], which clusters proteins by their sequence and structural similarities ensuring that its entries are evolutionary and structurally non-redundant. We further filter SCOPe by the length of up to 128 residues, which results in 3938 proteins.

We compare GAFL with FrameFlow and models for which we leave out all proposed architectural changes or only higher order message passing, respectively, while scaling the width of the layers such that the number of parameters remains the same. All models are trained with three different random seeds for 6500 epochs on one NVIDIA A100-80GB GPU, which takes around 6 days, and evaluated by generating backbones for lengths between 60 and 128 as in [67].

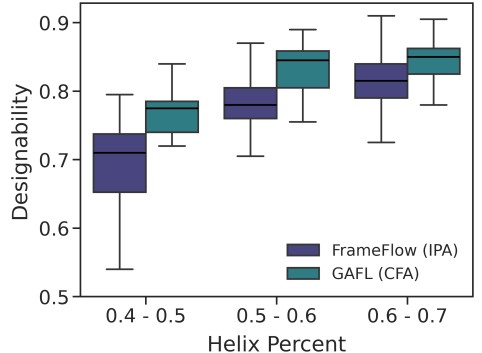

We find that GAFL's designability of 90.5% is 8 percentage points higher than that of the published FrameFlow model [67] trained on SCOPe (81.2%) as depicted in Table 3. If retrained with GAFL's training procedure, FrameFlow's architecture with original IPA achieves a designability of 88.2%. Using PGA-valued features and higher order message passing as proposed in CFA increases the designability by 1.4 and 2.3 percentage points, respectively. While the training procedure has a larger effect on the des-

Figure 4: Helix content and designabilities of 90 model checkpoints sampled during three training runs on the PDB dataset for GAFL and retrained FrameFlow, respectively. For each checkpoint, we sample 40 backbones per length in $\{100, 150, \ldots, 300\}$.

ignability, the improvement due to CFA can be considered to be significant since it persists across different random seeds. This can also be observed in the distribution of designabilities of 30 checkpoints sampled during the training procedure (Figure A.10) described in Section 4.1.

For the much larger PDB dataset, we conduct a small ablation study, in which we compare the performance of GAFL with CFA and retrained FrameFlow with original IPA across three different training runs, respectively (Table 2). Since GAFL consistently achieves higher designabilities, we can validate that the trend observed on SCOPe also holds true for training on the PDB.

Furthermore, we observe a correlation between over-representing $\alpha$-helices and achieving high designability (Figure 4) for 90 checkpoints sampled for each model during checkpoint selection

Table 2: Ablation of GAFL and FrameFlow on the PDB dataset. We report the results of three training runs and the published FrameFlow model.

| Method | Designability ($\uparrow$) | Diversity ($\downarrow$) |
|---|---|---|
| GAFL | $87.8^{87.9}_{84.3}$ | $0.35^{0.36}_{0.34}$ |
| FrameFlow | $84.4^{84.7}_{78.1}$ | $0.35^{0.35}_{0.34}$ |
| FrameFlow* | 84.6 | 0.35 |

*Published model weights

as described in Section 4.1. Crucially, GAFL checkpoints consistently show better designability in the low helix-content regime, which we can attribute to using CFA instead of IPA.

Table 3: Ablation of GAFL models with different elements of the proposed CFA architecture, trained on the SCOPe-128 dataset. For each model, we report the performance of three training runs with different random seeds, evaluated by generating 10 backbones for each length in $\{60, 61, \ldots, 128\}$.

| PGA | Higher Ord. Msg. | Checkp. Selection | Designability [%] ($\uparrow$) | Diversity ($\downarrow$) |
|:---:|:---:|:---:|:---:|:---:|
| ✓ | ✓ | ✓ | $90.5^{90.6}_{89.6}$ | $0.38^{0.39}_{0.36}$ |
| ✓ | | ✓ | $89.6^{90.3}_{89.0}$ | $0.37^{0.40}_{0.37}$ |
| | | ✓ | $88.2^{88.6}_{86.7}$ | $0.38^{0.39}_{0.38}$ |
| FrameFlow-SCOPe | | | 81.2 | 0.37 |

## 4.6 Discussion

Our results suggest that GAFL is one of the current state-of-the-art models for unconditional protein structure generation. GAFL outperforms the widely used, pre-trained model RFdiffusion in terms of diversity, novelty and inference time and is on par at designability. Remarkably, GAFL can achieve this performance without requiring pre-trained model weights while other non-pre-trained models often lack designability or show a mode-collapse towards generating helical structures.

Especially for generating small, highly designable backbones with distinct secondary structures, GAFL performs well, in particular better than RFdiffusion. Since in most protein design campaigns the goal is to incorporate the desired functionality into the smallest protein possible while exploring a large structural space, we consider this advantage of GAFL to be highly relevant for future developments and real-world applications of generative models for proteins.

Our ablation studies provide evidence that achieving high designability without over-representing $\alpha$-helices can be attributed to the replacement of IPA by the proposed CFA architecture. Since IPA is used in many current state-of-the-art architectures for backbone structure, CFA has the potential to enable improvements in many protein-related tasks.

**Limitations** While the proposed method GAFL achieves state-of-the-art performance in protein backbone generation, there is still room for improvement. We note that achieving high designability without compromising the diversity of secondary structures remains a grand challenge. GAFL does not perfectly capture the secondary structure distribution of natural proteins, especially for large proteins. Further, unconditional protein generation can be regarded as suitable benchmarking task for protein design models, however, most applications require conditional sampling. GAFL can be readily incorporated into existing frameworks for conditioning, e.g. on motifs [69] or symmetry [63].

## 5 Conclusion

We introduced Geometric Algebra Flow Matching (GAFL), a flow matching model for protein design based on FrameFlow [67]. GAFL relies on the proposed Clifford Frame Attention (CFA), an extension of the invariant point attention block from AlphaFold2 [32], by representing residue frames and geometric features of a protein in the projective geometric algebra. This enables the usage of the bilinear operations in the algebra to construct geometrically expressive messages between residues. The experiments demonstrate that the resulting model is state-of-the-art in the combination of the established metrics designability, diversity and novelty and performs notably well at generating designable small backbones with distinct secondary structures, containing $\beta$-strands in particular. Given the promising results of the extension of invariant point attention with geometric algebra presented in this work, we look forward to exploring its benefits for other protein-related tasks.

**Acknowledgments** This study received funding from the Klaus Tschira Stiftung gGmbH (HITS Lab). We acknowledge the National Academic Infrastructure for Supercomputing in Sweden (NAISS), partially funded by the Swedish Research Council through grant agreement no. 2021-29 for awarding this project access to the Berzelius resource provided by the Knut and Alice Wallenberg Foundation at the National Supercomputer Centre. The authors acknowledge support by the state of Baden-Württemberg through bwHPC and the German Research Foundation (DFG) through grant INST 35/1597-1 FUGG.

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

# A   Appendix

## A.1   Background

### A.1.1   Background on Geometric Algebra

In this section we give a short introduction to *Clifford algebra*, loosely following the presentation of Dorst et al. [26], Dorst and De Keninck [25] and Doran and Lasenby [24].

**General construction of Clifford algebras**

A Clifford algebra can be constructed from a vector space $V$ by extending it with an additional bilinear operation called the *geometric product*. We write this algebra as $\mathbb{G}(V)$. Elements of the algebra $\mathbf{A} \in \mathbb{G}(V)$ are called *multivectors* and are written in bold. In cases where we want to highlight that an algebra element coincides with a vector in $V$, we write it with a (bold) lower case letter. Other algebra elements are generally denoted by upper case letters. The geometric product has to fulfill the following properties [24]:

> **Definition A.1: Properties of the geometric product**
> 1. Associativity: $(\mathbf{A}\mathbf{B})\mathbf{C} = \mathbf{A}(\mathbf{B}\mathbf{C})$ $\hspace{3cm}$ $\mathbf{A}, \mathbf{B}, \mathbf{C} \in \mathbb{G}(V)$
> 2. Distributivity: $\mathbf{A}(\mathbf{B} + \mathbf{C}) = \mathbf{A}\mathbf{B} + \mathbf{A}\mathbf{C}$
> 3. Vectors square to scalars: $\mathbf{a}\mathbf{a} = \mathbf{a}^2 \in \mathbb{R}$

We denote the geometric product as juxtaposition of elements in order to distinguish it from other products liker inner and outer product.

**Algebra basis**   Given a basis of the underlying vector space $V$, $\{\mathbf{e}_i\}_{i=0}^{n}$, where $n$ is the dimension of the vector space, we can use the geometric product to construct a basis for the algebra.

$$\{1, \mathbf{e}_1, \ldots, \mathbf{e}_n, \mathbf{e}_1\mathbf{e}_2, \mathbf{e}_1\mathbf{e}_3, \ldots, \mathbf{e}_1 \ldots \mathbf{e}_n\}$$

In general, a geometric algebra over an $n$ dimensional vector space will have $2^n$ basis vectors. A general multivector can be written as a linear combination of these basis vectors. The basis elements can be further categorized into so called *grades* according to the number of basis vectors they contain, i.e. 1 will be of grade 0, $\mathbf{e}_i$ of grade 1, $\mathbf{e}_i\mathbf{e}_j$ of grade 2 and so on. With this definition it is possible to define the projection of an arbitrary multivector onto a specific grade [26].

> **Definition A.2: Grade projection**
> Let $\mathbb{G}^{[k]}(V)$ be the subspace of $\mathbb{G}(V)$ spanned by all multivectors of grade $k$. We then define the grade projection operator
> $$\langle \cdot \rangle_k : \mathbb{G}(V) \rightarrow \mathbb{G}^{[k]}(V), \tag{13}$$
> which selects the $k$-th grade of a given multivector.

Multivectors that contain only elements of one grade receive specific names according to their grade. Elements of grade 1 are *vectors*, elements of grade 2 are *bivectors*, elements of grade 3 are *trivectors* and so on.

**The metric**   So far we have only specified that the square of a vector should yield a scalar. In order to uniquely define a geometric algebra it is important to specify the exact values of these scalars, which will define a metric on the algebra. The choice of metric is crucial for the properties of the algebra and the geometric interpretation of its elements. For an algebra over an $n$-dimensional vector space we may choose $n$ scalars, one for each basis vector. It is common to work with a metric that assigns 1 or -1 to all non-zero values. This convention leaves $\{-1, 0, 1\}$ as possible choices for the scalars. One defines $\mathbb{G}_{n,m,l}$ as the geometric algebra with $n$ basis vectors squaring to 1, $m$ basis

vectors squaring to $-1$ and $l$ basis vectors squaring to $0$. Having fixed a metric for the vectors of the algebra we can go on to discuss the construction of a norm for general multivectors. The definition of the norm in geometric algebra uses the concept of *reversion* which is defined as follows:

**Definition A.3: Reversion**

Let $\mathbf{a}_1, \mathbf{a}_2, \ldots \mathbf{a}_n \in \mathbb{G}^{[1]}(V)$, $n \in \mathbb{N}$ be vectors and $\mathbf{A} = \prod_{i=1}^{n} \mathbf{a}_i$ the geometric product of these vectors, then the reverse of $\mathbf{A}$ is defined as

$$\tilde{\mathbf{A}} = \prod_{i=1}^{n} \mathbf{a}_{n-i+1}, \tag{14}$$

i.e. we reverse the order of the vector elements. For a general multivector, which may consist of a sum of elements of the form of 14, we apply the reverse operation to each summand individually.

A norm on general multtivectors can then be defined as:

**Definition A.4: Norm**

Let $\mathbf{A} \in \mathbb{G}(V)$ be a multivector, then the norm of $\mathbf{A}$ is defined as

$$\|\mathbf{A}\| = \sqrt{\langle \tilde{\mathbf{A}} \mathbf{A} \rangle_0}. \tag{15}$$

**The Euclidean geometric algebra $\mathbb{G}_3$**

One of the most prominent examples for a geometric algebra is the Euclidean geometric algebra $\mathbb{G}_3$, which is the geometric algebra over $\mathbb{R}^3$ with the standard Euclidean metric, i.e. a vector basis $\{\mathbf{e}_1, \mathbf{e}_2, \mathbf{e}_3\}$ fulfilling $\mathbf{e}_1^2 = \mathbf{e}_2^2 = \mathbf{e}_3^2 = 1$. Apart from the geometric product we can define two additional operations that will be useful for further analysis.

**Definition A.5: Inner and outer product**

Given two arbitrary vectors $\mathbf{a}, \mathbf{b} \in \mathbb{G}_3^{[1]}$ we can separate their geometric product into a symmetric and antisymmetric part

$$\mathbf{ab} = \frac{1}{2} (\mathbf{ab} + \mathbf{ba}) + \frac{1}{2} (\mathbf{ab} - \mathbf{ba}). \tag{16}$$

One typically refers to $\frac{1}{2} (\mathbf{ab} + \mathbf{ba}) =: \mathbf{a} \cdot \mathbf{b}$ as the **inner product** and $\frac{1}{2} (\mathbf{ab} - \mathbf{ba}) =: \mathbf{a} \wedge \mathbf{b}$ as the **outer product** of the two vectors. The geometric product of $\mathbf{a}$ and $\mathbf{b}$ can then be written as

$$\mathbf{ab} = \mathbf{a} \cdot \mathbf{b} + \mathbf{a} \wedge \mathbf{b}. \tag{17}$$

This definition only holds for vectors. For an extension to arbitrary multivectors see [24]. To further investigate the properties of these products we can look at the square of the sum of two vectors $(\mathbf{a} + \mathbf{b})^2$. Expanding the product and rearranging the terms yields

$$\frac{1}{2} (\mathbf{ab} + \mathbf{ba}) = \frac{1}{2} (\mathbf{a} + \mathbf{b})^2 - \mathbf{a}^2 - \mathbf{b}^2 = \mathbf{a} \cdot \mathbf{b}. \tag{18}$$

Since all terms on the right side are scalars due to the third property in Definition A.1, we can conclude that the inner product of two vectors is a scalar itself. In fact, 18 corresponds to the well known polarization identity from linear algebra, and since we defined the norm of vectors to be the standard Euclidean norm, the inner product from Definition A.5 is the standard Euclidean inner product.

Using 17 we can also introduce the notion of an orthogonal basis. Just as in linear algebra a vector basis $\{\mathbf{e}_i\}_i$ is called *orthogonal* if $\mathbf{e}_i \cdot \mathbf{e}_j = 0$ for $i \neq j$. In the following discussion we will

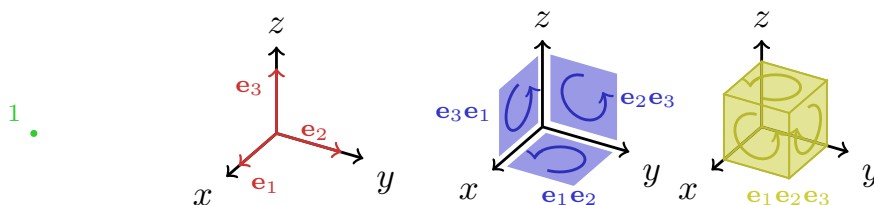

Figure A.1: Visualization of the different geometric primitives in $\mathbb{G}_3$. Vectors are directed line segments, bivectors are oriented areas and trivectors are oriented volumes. Orientations are indicated by arrows that have the same sense of rotation as the corresponding basis vectors when linked together end to tip.

always assume an orthogonal basis. As a consequence of 17 for $i \neq j$ basis elements anticommute $\mathbf{e}_i\mathbf{e}_j = -\mathbf{e}_j\mathbf{e}_i$ and the geometric product is equal to the outer product $\mathbf{e}_i\mathbf{e}_j = \mathbf{e}_i \wedge \mathbf{e}_j$.

**Geometric interpretation of the algebra elements**  As the name *geometric* algebra suggests, it is possible to assign geometric meaning to the algebra elements with grade greater than 0. Grade one elements or *vectors* keep their usual geometric interpretation as directed line segments. To find an interpretation for higher grade elements, one can observe that the outer product of two vectors $\mathbf{A} = \mathbf{a} \wedge \mathbf{b}$ defines a homogeneous subspace in the sense that for every vector in the span of $\mathbf{a}$ and $\mathbf{b}$ the following equation holds

$$\mathbf{x} \in \mathcal{A} = \text{span}\{\mathbf{a}, \mathbf{b}\} \iff \mathbf{x} \wedge \mathbf{A} = 0, \tag{19}$$

as shown in [26]. Additionally $\mathbf{A}$ has a magnitude according to Definition A.4 and also an orientation due to the antisymmetry of the outer product. For example $\mathbf{e}_1 \wedge \mathbf{e}_2$ has opposite orientation compared to $\mathbf{e}_2 \wedge \mathbf{e}_1 = -\mathbf{e}_1 \wedge \mathbf{e}_2$ as indicated by the relative minus sign. 2-*blades*, that is multivectors which can be written purely as the outer product of two vectors, can thus be interpreted as oriented areas, which lie within the subspace spanned by their generating vectors and area equal to the norm of the blade. Analogously 3-blades, multivectors equal to the outer product of three vectors, correspond to oriented volumes. Notably like vectors, 2-blades and 3-blades neither have a position in space nor do they have a specified shape. The only fixed properties are the magnitude of their area/volume as well as their orientation. The geometric interpretation of algebra elements is visualized in A.1.

Grades of the same dimensionality are closely related by a relation called *duality*. Graphically speaking, a plane in 3D space for example can be represented both as the plane itself or by the normal vector that is orthogonal to it. In algebraic terms this translates to e.g. $\mathbf{e}_3$ representing the normal vector of the plane described by the bivector $\mathbf{e}_1\mathbf{e}_2$. One says that these two elements are *dual* to each other. In the same way, scalars are dual to trivectors since trivectors, like scalars, have only one degree of freedom in three dimensions and are thus also sometimes called pseudoscalars. Formally duality in $\mathbb{G}_3$ can be defined as follows:

> **Definition A.6: Duality**
> Let $\mathbf{A} \in \mathbb{G}_3$ and $\mathbf{I} = \mathbf{e}_1\mathbf{e}_2\mathbf{e}_3$. Then the dual of $\mathbf{A}$ is given by
>
> $$\mathbf{A}^* = \mathbf{I}\mathbf{A} \tag{20}$$

Finally, it is also insightful to calculate the magnitude of the area represented by a 2-blade. To this end we notice that, using the Einstein sum convention, the outer product of two vectors $\mathbf{a} = a_i\mathbf{e}_i$, $\mathbf{b} = a_j\mathbf{e}_j$ can be written as

$$\mathbf{a} \wedge \mathbf{b} = a_ia_j\mathbf{e}_i \wedge \mathbf{e}_j = a_ia_j\mathbf{e}_i\mathbf{e}_j = \epsilon_{ijk}a_ib_j\mathbf{I}\mathbf{e}_k = \mathbf{I}(\mathbf{a} \times \mathbf{b}), \tag{21}$$

where $\epsilon_{ijk}$ is the Levi-Civita symbol and $\times$ denotes the usual vector cross product. This shows that in $\mathbb{G}_3$ the outer product yields the 2-blade which is dual to the resulting vector of the cross product

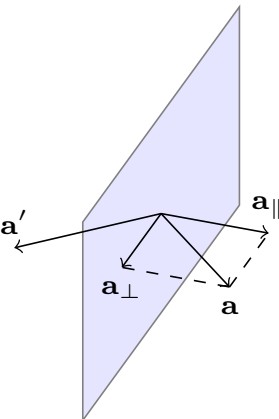

Figure A.2: Visualization of the reflection of a vector $\mathbf{a}$ at a plane.

with the same magnitude. In fact, the outer product can be seen as generalization of the cross product to arbitrary dimensions. The magnitude can be calculated from 15, yielding

$$\|\mathbf{a} \wedge \mathbf{b}\| = \sqrt{\langle \|\mathbf{a} \times \mathbf{b}\|^2 \tilde{\mathbf{II}} \rangle_0} = \sqrt{\|\mathbf{a} \times \mathbf{b}\|^2} = \|\mathbf{a}\|\|\mathbf{b}\||\sin(\alpha)|, \qquad (22)$$

where $\alpha$ is the angle enclosed by $\mathbf{a}$ and $\mathbf{b}$. This is exactly the area of the parallelogram spanned by the two vectors. Similar calculations can be performed for the case of 3-blades to show that their magnitude is equal to the volume of the parallelepiped spanned by their three generating vectors. These findings further support the idea of interpreting bivectors and trivectors as area and volume elements respectively.

**Orthogonal transformations**  Another important class of operations which geometric algebra can describe in an efficient way are orthogonal transformations, that is transformations $T : \mathbb{G}_3 \to \mathbb{G}_3$ which preserve the inner product between vectors. In group theoretic terms, these transformations form the group $O(3)$. In order to construct such transformations we first look at how reflections are handled in $\mathbb{G}_3$.

> **Definition A.7: Reflection**
> Let $\mathbf{a} \in \mathbb{G}_3^{[1]}$ be a vector and $\mathbf{n} \in \mathbb{G}_3^{[1]}$ be the unit normal vector of a plane. The reflection of $\mathbf{a}$ at that plane is then given by
>
> $$\mathbf{a}' = -\mathbf{n}\mathbf{a}\mathbf{n}. \qquad (23)$$

To see why this indeed corresponds to a reflection, we decompose $\mathbf{a}$ into a parallel and orthogonal part with respect to the normal vector $\mathbf{n}$ as shown in A.2. Algebraically this can be written as

$$\mathbf{a} = \mathbf{n}^2\mathbf{a} = \mathbf{n}(\mathbf{n}\mathbf{a}) = \mathbf{n}(\mathbf{n} \cdot \mathbf{a} + \mathbf{n} \wedge \mathbf{a}) \qquad (24)$$

$$= \underbrace{\mathbf{n}(\mathbf{n} \cdot \mathbf{a})}_{=\mathbf{a}_\parallel} + \underbrace{\mathbf{n}(\mathbf{n} \wedge \mathbf{a})}_{=\mathbf{a}_\perp}. \qquad (25)$$

The first summand corresponds to the usual projection formula from linear algebra and can thus be identified as the parallel part. Since the whole expression equals $\mathbf{a}$, the second summand must be the orthogonal part. The reflected vector can now be obtained by reversing the sign of $\mathbf{a}_\parallel$.

$$\mathbf{a}' = -\mathbf{a}_\parallel + \mathbf{a}_\perp = -\mathbf{n}(\mathbf{a} \cdot \mathbf{n}) + \mathbf{n}(\mathbf{n} \wedge \mathbf{a}) \qquad (26)$$

$$= -\mathbf{n}(\mathbf{a} \cdot \mathbf{n}) - \mathbf{n}(\mathbf{a} \wedge \mathbf{n}) = -\mathbf{n}(\mathbf{a} \cdot \mathbf{n} + \mathbf{a} \wedge \mathbf{n}) \qquad (27)$$

$$= -\mathbf{n}\mathbf{a}\mathbf{n}. \qquad (28)$$

From 26 to 27, we used the antisymmetry of the outer product. This construction of reflections shows that elements of Euclidean geometric algebra can simultaneously be interpreted as geometric objects as well as transformation operators, a concept which also translates to other geometric algebras.

To construct general orthogonal transformations we can make use of the following theorem.

> **Theorem A.1: Cartan-Dieudonné theorem**
> Let $(V, q)$ be a nondegenerate space of dimension $n$, then any orthogonal transformation can be written as a composition of at most $n$ reflections.

For a proof see [41]. In the context of $\mathbb{G}_3$, this implies that any orthogonal transformation of a vector, which in addition to reflections comprises rotations around lines through the origin as well as compositions of both operations, can be written as

$$\mathbf{a}' = \pm \mathbf{V} \mathbf{a} \widetilde{\mathbf{V}}, \ \mathbf{V} \in \mathbb{G}_3, \ \|\mathbf{V}\| = 1. \tag{29}$$

The definition can be extended to higher grade blades, by transforming each of its vectors individually. The transformation of a general blade can then be written as

$$\bigwedge_{i=1}^{k} \mathbf{a}_i \rightarrow \bigwedge_{i=1}^{k} \left( \pm \mathbf{V} \mathbf{a}_i \widetilde{\mathbf{V}} \right) = (\pm 1)^k \mathbf{V} \left( \bigwedge_{i=1}^{k} \mathbf{a}_i \right) \widetilde{\mathbf{V}}. \tag{30}$$

The property that the outer product of the transformed vectors is equal to the transformed outer product makes any orthogonal transformation a so called *outermorphism*. As a consequence, orthogonal transformations preserve the grade of blades, which geometrically translates to the fact that vectors are transformed to vectors, bivectors to bivectors and so on, which is what we would expect from a geometric transformation (a vector will not become a volume when rotated). A formal proof of this property can be found in [26].

The multivector $\mathbf{V}$ in 29 is called a *versor*. Versors together with the geometric product form a group on their own, the so called $\mathrm{Pin}(3)$ group. This group is said to be the *double cover* of $O(3)$ (see [40]). 30 is both a representation of $O(3)$ and $\mathrm{Pin}(3)$. One can generalize this definition to arbitrary geometric algebras in the following way:

> **Definition A.8: The Pin(n,m,l) and Spin(n,m,l) groups**
> Let $\mathcal{V} = \{v \in \mathbb{G}_{n,m,l} \,|\, \|\mathbf{v}\| = 1\}$ be the set of versors . Then $\mathcal{V}$ together with the geometric product forms the group $\mathrm{Pin}(n, m, l)$.
> Let $\mathcal{V} = \{v \in \mathbb{G}_{n,m,l}^+ \,|\, \|\mathbf{v}\| = 1\}$ be the set of even versors, i.e. versors that only involve even grades. Then $\mathcal{V}$ together with the geometric product forms the group $\mathrm{Spin}(n, m, l)$.

**The projective geometric algebra $\mathbb{G}_{3,0,1}$**
The Euclidean geometric algebra $\mathbb{G}_3$ allows for a powerful description of geometric objects in 3D space. However, it lacks the ability to describe absolute positions. For example, planes parametrized by 2-blades are restricted to go through the origin, and vectors only describe a relative displacement, not points in space. These problems are solved by the projective geometric algebra (PGA), which is the geometric algebra $\mathbb{G}_{3,0,1}$ with three *Euclidean* basis vectors squaring to 1, $\mathbf{e}_1^2 = \mathbf{e}_2^2 = \mathbf{e}_3^2 = 1$ and one *null* vector squaring to 0, $\mathbf{e}_0^2 = 0$. We thus use an algebra based on a four dimensional vector space to describe 3D space, similarly to what one does with homogeneous coordinates. In the following we will denote PGA vectors which only contain a Euclidean part by $\vec{\mathbf{a}}$.

There are multiple ways to interpret the different algebraic elements geometrically. Here, we will focus on the plane-based approach as described in [25], since this yields the most useful description of Euclidean motions as orthogonal transformations. Elements of grade 1 are interpreted not as points but as planes, i.e. a vector of the form

$$\mathbf{n} = \vec{\mathbf{n}} + \delta \mathbf{e}_0 \tag{31}$$

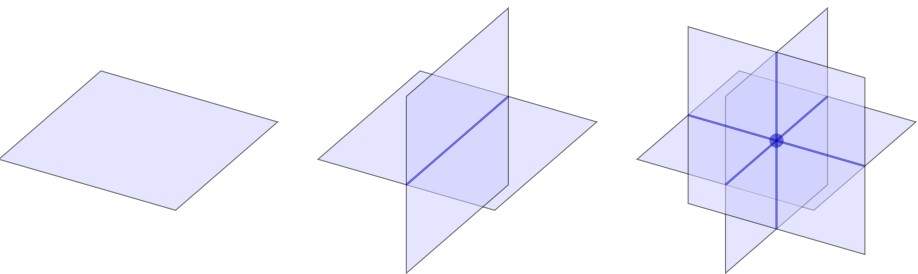

Figure A.3: Visualization of the different geometric primitives in $\mathbb{G}_{3,0,1}$. Vectors represent planes, 2-blades represent lines resulting from the intersecting of its generating planes and 3-blades represent points as the intersection of three planes.

Table A.1: Basis elements of the projective geometric algebra along with their geometric interpretation

| Grade | Basis Elements | Geometric Interpretation |
|---|---|---|
| 0 | $\{1\}$ | scalar |
| 1 | $\{\mathbf{e}_0, \mathbf{e}_1, \mathbf{e}_2, \mathbf{e}_3\}$ | planes |
| 2 | $\{\mathbf{e}_0\mathbf{e}_1, \mathbf{e}_0\mathbf{e}_2, \mathbf{e}_0\mathbf{e}_3, \mathbf{e}_1\mathbf{e}_2, \mathbf{e}_1\mathbf{e}_3, \mathbf{e}_2\mathbf{e}_3\}$ | lines, vanishing lines |
| 3 | $\{\mathbf{e}_0\mathbf{e}_1\mathbf{e}_2, \mathbf{e}_0\mathbf{e}_1\mathbf{e}_3, \mathbf{e}_0\mathbf{e}_2\mathbf{e}_3, \mathbf{e}_1\mathbf{e}_2\mathbf{e}_3\}$ | points, vanishing points |
| 4 | $\{\mathbf{e}_0\mathbf{e}_1\mathbf{e}_2\mathbf{e}_3\}$ | pseudoscalar |

corresponds to a plane with (Euclidean) normal vector $\vec{\mathbf{n}}$ and a distance $\delta$ from the origin. In turn, the Euclidean vectors $\mathbf{e}_1, \mathbf{e}_2, \mathbf{e}_3$ correspond to basis planes through the origin, whereas $\mathbf{e}_0$ is interpreted as the plane at infinity. Notably any multiple of 31 of the form $\alpha(\vec{\mathbf{n}} + \delta\mathbf{e}_0)$, $\alpha \in \mathbb{R}$ represents the same plane.

Similarly to the case of $\mathbb{G}_3$, we can make use of 19 to obtain a geometric interpretation for blades of higher grade. For 2-blades of the form $\mathbf{L} = \mathbf{a} \wedge \mathbf{b}$, the subspace spanned by $\mathbf{a}$ and $\mathbf{b}$ corresponds to all planes which contain the line of intersection between the two original planes. It is thus sensible to take $\mathbf{L}$ as representation of this line. Similarly, a 3-blade $\mathbf{X} = \mathbf{a} \wedge \mathbf{b} \wedge \mathbf{c}$ corresponds to the subspace spanned by all planes which contain the point of intersection of $\mathbf{a}, \mathbf{b}$, and $\mathbf{c}$, and thus 3-blades represent points in PGA. The logic behind this construction is visualized in A.3.

PGA also contains additional classes of geometric objects which e.g. result from taking the outer product of parallel planes. For a further discussion of these, we refer to [25].

**Meet and Join**  We have seen that the outer product in PGA can be used to calculate the intersection of geometric objects. In this context it is thus also known as the so called *meet*. There is also an opposite operation called the *join*[3], which, as the name suggests, e.g. maps two points to the line which passes through both of them. In order to properly define the join we first extend the concept of duality from $\mathbb{G}_3$ to $\mathbb{G}_{3,0,1}$.

> **Definition A.9: Hodge duality**
> Let $\mathbf{X} \in \mathbb{G}_{3,0,1}$ be a blade. Then its *hodge dual* $\star\mathbf{X}$ is defined via
>
> $$\mathbf{X} \star\mathbf{X} = \left(\mathbf{X}_e \widetilde{\mathbf{X}_e}\right) \mathcal{I}, \tag{32}$$
>
> where $\mathbf{X}_e$ is the Euclidean part of $\mathbf{X}$ without $\mathbf{e}_0$. For general multivectors the hodge dual is performed bladewise.

As explained in [25], this definition is slightly different compared to Definition A.6 since in PGA $\mathbf{e}_0$ does not have a multiplicative inverse. However, the overall idea to map between the different

---
[3]In some literature the roles of meet and join are actually reversed, i.e. the outer product takes the role of the join. This seeming contradiction can be explained by the fact that we follow the approach of interpreting vectors as planes and not as points.

subspaces of equal dimensionality is still the same. In practice the hodge dual maps a basis element simply to the element which contains all the basis vectors not present in the initial multivector (in some situations with an additional minus sign), e.g. $\star\mathbf{e}_0\mathbf{e}_3 = \mathbf{e}_1\mathbf{e}_2$ (confer [25] for more details). Using the hodge dual one can define the *join* as follows.

---

**Definition A.10: Join**
Let $\mathbf{A}, \mathbf{B} \in \mathbb{G}_{3,0,1}$, then the join between these multivectors is defined as

$$\mathbf{A} \vee \mathbf{B} = \star^{-1}(\star\mathbf{B} \wedge \star\mathbf{A}), \tag{33}$$

where $\star^{-1}$ is the inverse hodge dual, which differs from the usual hodge dual by a sign for some elements.

---

As mentioned earlier, the join can be viewed as being dual to the meet, linking geometric objects together instead of finding their incidence. In that way, the join of two points is the line connecting both of them and the join of three points results in a plane containing all three points.

In PGA a norm for multivectors can be defined analogously to Definition A.4. However, since PGA contains the null vector $\mathbf{e}_0$, only half of the components of a general multivector contribute to the value of the norm. It is thus useful to define a second norm which depends on all of the components which contain $\mathbf{e}_0$.

---

**Definition A.11: Infinity norm**
Let $\mathbf{A} \in \mathbb{G}_{3,0,1}$, then its *infinity norm* is given by

$$\|\mathbf{A}\|_\infty = \|\star\mathbf{A}\|. \tag{34}$$

---

**Euclidean transformations**  Similarly to the case of $\mathbb{G}_3$, one can write the reflection of a plane $\mathbf{m}$ in another plane $\mathbf{p}$ as

$$\mathbf{m}' = -\mathbf{p}\mathbf{m}\mathbf{p}, \ \|\mathbf{p}\| = 1. \tag{35}$$

Reflection of higher grade blades, e.g. lines and points, can again be achieved by reflecting each vector in the blade individually and making use of the fact that the reflection is an outermorphism

$$\bigwedge_{i=1}^k \mathbf{a}_i \rightarrow \bigwedge_{i=1}^k (-\mathbf{p}\mathbf{a}_i\mathbf{p}) = (-1)^k \mathbf{p} \left( \bigwedge_{i=1}^k \mathbf{a}_i \right) \mathbf{p}. \tag{36}$$

The crucial difference in comparison to $\mathbb{G}_3$ is that the reflecting plane is no longer restricted to go through the origin. This allows one to do consecutive reflections not only in intersecting planes, but also in parallel planes. Two important special cases are the reflections in two intersecting planes and the reflecions in two parallel planes as shwon in A.4. The first case corresponds to a rotation around the line of intersection by an angle which is twice the angle enclosed by the planes. The second case results in a translation along the distance vector by an amount equal to twice the distance between the planes (see [25]).

Compositions of these two types of transformations make up the special Euclidean group SE(3), which can be seen as the group of rigid body motions. This means that every Euclidean transformation which does not involve reflections can be embedded as an element of the even subalgebra $\mathbb{G}_{3,0,1}^+ = \{A \in \mathbb{G}_{3,0,1} \,|\, \langle A \rangle_k = 0, \ k \text{ odd}\}$, as stated in the following theorem:

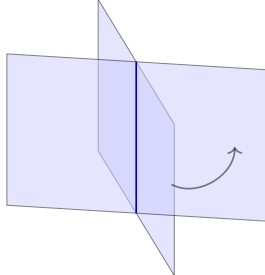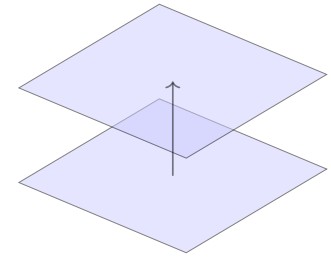

Figure A.4: Visualization of the different geometric transformations in $\mathbb{G}_{3,0,1}$

**Theorem A.2: Euclidean motors**
Let $T \in \mathrm{SE}(3)$ be an element of the special Euclidean group. Then there exists a multivector called *motor* in the even subalgebra $\mathbf{M} \in \mathbb{G}_{3,0,1}^+$ with $\|\mathbf{M}\| = 1$ such that

$$\mathbf{X}' = \mathbf{M}\mathbf{X}\widetilde{\mathbf{M}}, \tag{37}$$

with an arbitrary multivector $\mathbf{X} \in \mathbb{G}_{3,0,1}$, is a representation of $T$ on $\mathbb{G}_{3,0,1}$.

We emphasize that the expression in 37 can be applied to any geometric object which is representable in the algebra, i.e. it does not matter if $\mathbf{X}$ is a point, a vector, or a plane; the correct equation to transform it always takes the above form. This remarkable property can also be found for other important operations in PGA. In fact, we have already mentioned the *meet* operation which allows to calculate intersections of arbitrary geometric objects $\mathbf{A}, \mathbf{B} \in \mathbb{G}_{3,0,1}$ via the universal formula $\mathbf{A} \wedge \mathbf{B}$, as well as the *join* which can likewise be applied to pairs of arbitrary objects.

**Metric Relations**   We have already seen how to transform objects, find incidences between them, and join them together. Furthermore the basic operations of PGA allow to calculate a host of different metric relations between its elements. Below we provide a non-complete list of examples.

**Theorem A.3: Metric relations**
Let $\mathbf{p}_1, \mathbf{p}_2 \in \mathbb{G}_{3,0,1}^{[1]}$ be planes, $\mathbf{L} \in \mathbb{G}_{3,0,1}^{[2]}$ be a line and $\mathbf{P}_1, \mathbf{P}_2 \in \mathbb{G}_{3,0,1}^{[3]}$ be points, which are all normalized, i.e.

$$\|\mathbf{p}_1\| = \|\mathbf{p}_2\| = \|\mathbf{L}\| = \|\mathbf{P}_1\| = \|\mathbf{P}_2\| = 1,$$

then we can calculate the following relations:

- Distance between points $\mathbf{P}_1, \mathbf{P}_2$

$$\|\mathbf{P}_1 \vee \mathbf{P}_2\| \tag{38}$$

- Distance between point $\mathbf{P}_1$ and line $\mathbf{L}$

$$\|\mathbf{P}_1 \vee \mathbf{L}\| \tag{39}$$

- Distance between point $\mathbf{P}_1$ and plane $\mathbf{p}_1$

$$\|\mathbf{P}_1 \wedge \mathbf{p}_1\|_\infty \tag{40}$$

- Angle between planes $\mathbf{p}_1, \mathbf{p}_2$

$$\sin^{-1}\left(\|\mathbf{p}_1 \wedge \mathbf{p}_2\|\right) \tag{41}$$

- Angle between plane $\mathbf{p}_1$ and line $\mathbf{L}$

$$\sin^{-1}\left(\|\langle \mathbf{p}_1 \mathbf{L} \rangle_3\|\right) \tag{42}$$

For an extensive list of operations see [25].

## A.2 Methodology

### A.2.1 GAFL architecture

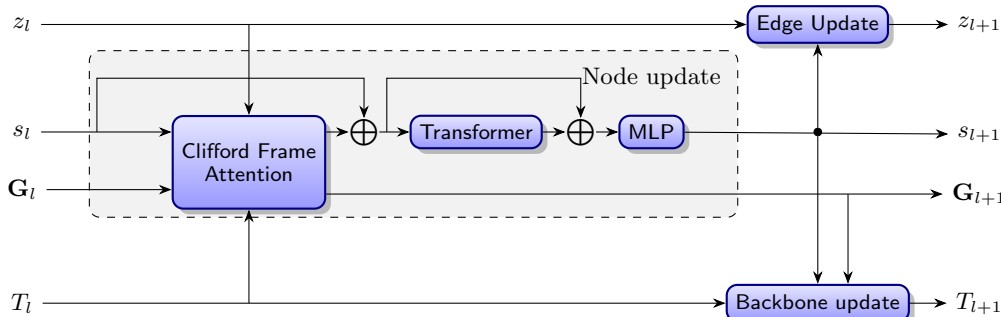

Figure A.5: High level overview over the GAFL architecture. The architecture was adapted from FrameDiff/FrameFlow [68, 67]. We replaced invariant point attention (IPA) with Clifford frame attention (CFA) and added geometric node features $\mathbf{G}$.

We adapted the GAFL architecture from FrameDiff/FrameFlow [68, 67] and replaced invariant point attention (IPA) with Clifford frame attention (CFA) (see Algorithm 3). Furthermore we introduced geometric node features $\mathbf{G}_i$ that are used in the prediction of geometric attention values and backbone frame updates. An overview over the architecture is shown in Figure A.5. The backbone update block is presented in Algorithm 6. For details on the edge update we refer to [68].

### A.2.2 Invariant Point Attention

In the following we provide a short overview over the IPA architecture.

Invariant point attention uses local frames to construct the messages between nodes, where the coordinate frames are given by the frames of the individual residues. The full procedure is presented in algorithm 1 and visualized in A.6. The part of IPA that processes geometric information (red nodes in A.6) can be summarized as follows:

1. Learn a certain number of *local* vector valued queries, keys and values $\vec{\mathbf{q}}_i, \vec{\mathbf{k}}_i, \vec{\mathbf{v}}_i$.

2. In the calculation of attention scores, transform the local queries and keys into the global frame via $T_i \circ \vec{\mathbf{q}}_i$ and calculate the squared distance between them. Importantly they should be thought of as points in 3D space rather than vectors, meaning that they change under

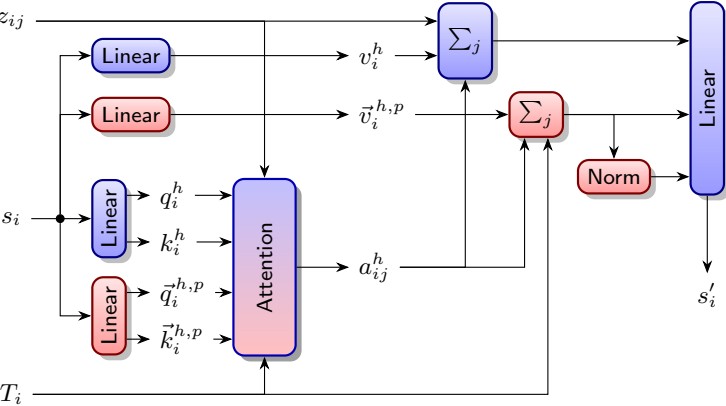

Figure A.6: High level overview of invariant point attention [32]. Blue nodes represent layers which process scalar information, while red nodes represent layers which process geometric information.

---

**Algorithm 1** Invariant point attention (IPA) [32]

1: **procedure** IPA($\{\mathbf{s}_i\}, \{\mathbf{z}_{ij}\}, \{T_i\}, N_{\text{head}} = 8, c = 128, N_{\text{query points}} = 8, N_{\text{point values}} = 12$)
2:      $\mathbf{q}_i^h, \mathbf{k}_i^h, \mathbf{v}_i^h = \text{LinearNoBias}(\mathbf{s}_i)$          $\triangleright \mathbf{q}_i^h, \mathbf{k}_i^h, \mathbf{v}_i^h \in \mathbb{R}^c, h \in \{1, \ldots, N_{\text{head}}\}$
3:      $\vec{\mathbf{q}}_i^{hp}, \vec{\mathbf{k}}_i^{hp} = \text{LinearNoBias}(\mathbf{s}_i)$        $\triangleright \vec{\mathbf{q}}_i^{hp}, \vec{\mathbf{k}}_i^{hp}, \in \mathbb{R}^3, p \in \{1, \ldots, N_{\text{query points}}\}$
4:      $\vec{\mathbf{v}}_i^{hp} = \text{LinearNoBias}(\mathbf{s}_i)$          $\triangleright \vec{\mathbf{v}}_i^{hp} \in \mathbb{R}^3, p \in \{1, \ldots, N_{\text{point values}}\}$
5:      $b_{ij}^h = \text{LinearNoBias}(\mathbf{z}_{ij})$
6:      $w_C = \sqrt{\frac{2}{9 N_{\text{query points}}}},$
7:      $w_L = \sqrt{\frac{1}{3}}$
8:      $a_{ij}^h = \text{softmax}_j \left( w_L \left( \frac{1}{\sqrt{c}} \mathbf{q}_i^{h\top} \mathbf{k}_j^h + b_{ij}^h - \frac{\gamma^h w_C}{2} \sum_p \left\| T_i \circ \vec{\mathbf{q}}_i^{hp} - T_j \circ \vec{\mathbf{k}}_j^{hp} \right\|^2 \right) \right)$
9:      $\tilde{\mathbf{o}}_i^h = \sum_j a_{ij}^h \mathbf{z}_{ij}$
10:     $\mathbf{o}_i^h = \sum_j a_{ij}^h \mathbf{v}_j^h$
11:     $\vec{\mathbf{o}}_i^{hp} = T_i^{-1} \circ \sum_j a_{ij}^h \left( T_j \circ \vec{\mathbf{v}}_j^{hp} \right)$
12:     $\tilde{\mathbf{s}}_i = \text{Linear} \left( \text{concat}_{h,p} \left( \tilde{\mathbf{o}}_i^h, \mathbf{o}_i^h, \vec{\mathbf{o}}_i^{hp}, \left\| \vec{\mathbf{o}}_i^{hp} \right\| \right) \right)$
13:     **return** $\{\tilde{\mathbf{s}}_i\}$
14: **end procedure**

---

**Algorithm 2** Original backbone update [32]

1: **procedure** BACKBONEUPDATE($\{\mathbf{s}_i\}, \{T_i\}$)
2:      $b_i, c_i, d_i, \vec{\mathbf{t}}_i = \text{Linear}(\mathbf{s}_i)$
3:      $(a_i, b_i, c_i, d_i) = (1, b_i, c_i, d_i) / \sqrt{1 + b_i^2 + c_i^2 + d_i^2}$
4:      $R_i = \text{QuaternionToMatrix}(a_i, b_i, c_i, d_i)$
5:      $\tilde{T}_i = (R_i, \vec{\mathbf{t}}_i)$
6:      **return** $T_i \circ \tilde{T}_i$
7: **end procedure**

---

        translations, which vectors would not, and also making the $L^2$-norm of their separation a somewhat natural map to construct invariant attention scores.

3. In the message passing step, transform vector valued features from the neighboring frame $j$ to the node frame $i$ via $T_i^{-1} \circ T_j \circ \vec{\mathbf{q}}_i$ and aggregate the vector valued information over all neighbors.

4. Finally, concatenate the output vectors along with their norm with the remaining scalar output features and put them through a final linear layer.

The update of the backbone frames, which is used in conjunction with IPA e.g. in [32, 67], is accomplished by learning an SE(3) transformation per frame which is concatenated with the current frame representation. To this end, the network predicts a quaternion for the rotation and a translation vector from the scalar node features $\mathbf{s}_i$ as shown in algorithm 2

### A.2.3 Clifford frame attention

An overview over the CFA architecture is given in Figure A.7. The proposed changes to the original IPA and FrameDiff architecture are shown in algorithms 3 and 6, highlighted in blue. They can be summarized as follows:

1. We replace point-valued attention values with multivectors $\mathbf{V}_i^{hp}$ and also introduce geometric node features $\mathbf{G}_i \in \mathbb{G}_{3,0,1}$.

2. We compute node features $\mathbf{T}^{\text{rel}}$ containing aggregated relative transformations between frames $\mathbf{T}_i$ and $\mathbf{T}_j$ as

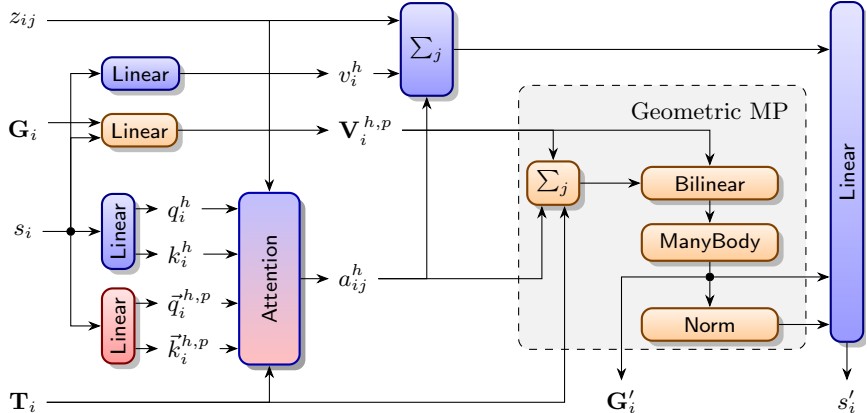

Figure A.7: Overview of Clifford frame attention. Blue nodes represent layers which process scalar information, red nodes represent layers which process point valued information and orange nodes represent layers which process features in the PGA. The central innovation is the novel construction of geometric messages, summarized in the grey box. To retain readability we omit the calculation of relative frame transformations in the chart.

$$\mathbf{T}_i^{\text{rel}} \equiv \sum_j a_{ij} \mathbf{T}_i^{-1} \mathbf{T}_j \tag{43}$$

These features are concatenated with the remaining geometric features and also passed directly to the backbone update block via a residual connection.

3. Messages are formed by applying geometric bilinear layers from [13] to node features of each node pair, as described in algorithm 4. The EquiLinear layer refers to the most general equivariant linear layer in PGA, also introduced in [13], which for input features $\mathbf{X}_n$ can be written as

$$\text{EquiLinear}(\mathbf{X}_n) = \sum_m \left[ \sum_{k=0}^4 w_{knm} \langle \mathbf{X}_m \rangle_k + \sum_{k=0}^3 v_{knm} \mathbf{e}_0 \langle \mathbf{X}_m \rangle_k \right]. \tag{44}$$

Although we do not need this property for enforcing equivariance, we use it for its parameter efficiency and geometric inductive bias.

4. We construct implicit higher body messages inspired by MACE [6]. Bilinearity of the geometric product and the join means that repeated products of aggregated two-body messages correspond to a sum of $N$-body messages as shown in eq. 11. In algorithm 5, we use the geometric product and the join to construct three-body messages from aggregated two-body messages, stored in node features $\mathbf{A}$.

5. In addition to the Euclidean norm we also compute the infinity norm (see Definition A.11) of multivector features after message passing.

6. In the Backbone update step (Algorithm 6), we concatenate scalar and geometric features along the feature dimension and pass them through an MLP to predict multivectors $\mathbf{R}$ and $\mathbf{S}$ that parameterize a rotation and translation respectively. These are then multiplied with the current frames to compute frame updates. The main difference to the backbone update as used in [32, 67] is the inclusion of geometric features and the use of the MLP. The prediction of rotor and translator instead of a rotation matrix and translation vector is just a matter of representation.

### A.2.4 Equivariance of GAFL

As in the original IPA formulation, SE(3) equivariance of the GAFL architecture is based on expressing geometric features in canonically induced local frames, as described in section 3.

More specifically, going through the architecture step by step, the attention scores are calculated using the $L_2$ norm of the difference of point features, which is E(3) invariant. Equivariance of the

---

**Algorithm 3** Clifford frame attention

---

1: **procedure** CFA($\{\mathbf{s}_i\}, \{\mathbf{G}_i\}, \{\mathbf{z}_{ij}\}, \{T_i\}, N_{\text{head}} = 8, c = 128, N_{\text{query points}} = 8, N_{\text{point values}} = 4$)

2:      $\mathbf{q}_i^h, \mathbf{k}_i^h, \mathbf{v}_i^h = \text{LinearNoBias}(\mathbf{s}_i)$                     $\triangleright \mathbf{q}_i^h, \mathbf{k}_i^h, \mathbf{v}_i^h \in \mathbb{R}^c, h \in \{1, \dots, N_{\text{head}}\}$

3:      $\vec{\mathbf{q}}_i^{hp}, \vec{\mathbf{k}}_i^{hp} = \text{LinearNoBias}(\mathbf{s}_i)$         $\triangleright \vec{\mathbf{q}}_i^{hp}, \vec{\mathbf{k}}_i^{hp}, \in \mathbb{R}^3, p \in \{1, \dots, N_{\text{query points}}\}$

4:      $\mathbf{V}_i^p = \text{Linear}(\mathbf{s}_i)$                            $\triangleright \mathbf{V}_i^p \in \mathbb{R}^{16}, p \in \{1, \dots, N_{\text{point values}}\}$

5:      $\mathbf{V}_i^{hp} = \text{EquiLinear}(\text{concat}_p(\mathbf{V}_i^p, \mathbf{G}_i^p))$

6:      $b_{ij}^h = \text{LinearNoBias}(\mathbf{z}_{ij})$

7:      $w_C = \sqrt{\frac{2}{9N_{\text{query points}}}},$

8:      $w_L = \sqrt{\frac{1}{3}}$

9:      $a_{ij}^h = \text{softmax}_j \left( w_L \left( \frac{1}{\sqrt{c}} \mathbf{q}_i^{h\top} \mathbf{k}_j^h + b_{ij}^h - \frac{\gamma^h w_C}{2} \sum_p \left\| T_i \circ \vec{\mathbf{q}}_i^{hp} - T_j \circ \vec{\mathbf{k}}_j^{hp} \right\|^2 \right) \right)$

10:     $\tilde{\mathbf{o}}_i^h = \sum_j a_{ij}^h \mathbf{z}_{ij}$

11:     $\mathbf{o}_i^h = \sum_j a_{ij}^h \mathbf{v}_j^h$

12:     $\mathbf{T}_i^{\text{rel } h} = \sum_j a_{ij}^h \mathbf{T}_i^{-1} \mathbf{T}_j$

13:     $\tilde{\mathbf{V}}_i^{hp} = \text{EquiLinear}\left( \text{concat}\left( \mathbf{V}_i^{hp}, \mathbf{T}_i^{\text{rel } h} \right) \right)$

14:     $\mathbf{O}_i^{hp} = \text{GeometricBilinear}\left( \mathbf{T}_i^{-1} \sum_j a_{ij} \left( \mathbf{T}_j \tilde{\mathbf{V}}_j^{hp} \mathbf{T}_j^{-1} \right) \mathbf{T}_i, \mathbf{V}_i^{hp} \right)$

15:     $\mathbf{O}_i^{hp} = \text{GeometricManyBodyProduct}(\mathbf{O}_i^{hp})$

16:     $\tilde{\mathbf{s}}_i = \text{Linear}\left( \text{concat}_{h,p}\left( \tilde{\mathbf{o}}_i^h, \mathbf{o}_i^h, \mathbf{O}_i^{hp}, \left\| \mathbf{O}_i^{hp} \right\|, \left\| \mathbf{O}_i^{hp} \right\|_\infty, \mathbf{T}_i^{\text{rel } h} \right) \right)$

17:     $\tilde{\mathbf{G}}_i = \text{EquiLinear}(\mathbf{O}_i^{hp})$

18:     **return** $\left\{ \tilde{\mathbf{s}}_i, \tilde{\mathbf{G}}_i, \mathbf{T}_i^{\text{rel}} \right\}$

19: **end procedure**

---

---

**Algorithm 4** GeometricBilinear [13]

---

1: **procedure** GEOMETRICBILINEAR($\mathbf{A}, \mathbf{B}$)

2:      $\mathbf{G}_L = \text{EquiLinear}(\mathbf{A})$

3:      $\mathbf{G}_R = \text{EquiLinear}(\mathbf{B})$

4:      $\mathbf{G} = \text{GeometricProduct}(\mathbf{G}_L, \mathbf{G}_R)$

5:      $\mathbf{J}_L = \text{EquiLinear}(\mathbf{A})$

6:      $\mathbf{J}_R = \text{EquiLinear}(\mathbf{B})$

7:      $\mathbf{J} = \text{Join}(\mathbf{J}_L, \mathbf{J}_R)$

8:      **return** $\text{EquiLinear}(\text{concat}(\mathbf{G}, \mathbf{J}))$

9: **end procedure**

---

message aggregation step is guaranteed by the expression

$$T_i^{-1} \circ T_j \circ \vec{v}_j^{hp}$$

in line 11 of algorithm 1, where $\vec{v}_j^{hp}$ are SE(3) invariant point values and the frames $\{T_i\}$ transform according to $T_i \to T_{global} \circ T_i$ such that the whole expression remains invariant:

$$T_i^{-1} \circ T_{global}^{-1} \circ T_{global} \circ T_j \circ \vec{v}_j^{hp} = T_i^{-1} \circ T_j \circ \vec{v}_j^{hp}.$$

In GAFL, we do not modify the calculation of attention scores from IPA, which means that their invariance remains ensured. During message passing, we use the same construction as above (see line 11 of Algorithm 3), but use a different representation of SE(3), namely multivector features instead of point features. The choice of representation, however, does not influence the invariance of the whole expression. Also the relative frame transformations $\mathbf{T}_i^{-1}\mathbf{T}_j$ we compute are invariant. All subsequent layers including the GeometricBilinear layer and the ManyBodyProduct layer operate exclusively on invariant node features, hence overall equivariance is retained throughout those layers.

---

**Algorithm 5** GeometricManyBodyProduct

1: **procedure** GEOMETRICMANYBODYPRODUCT($\mathbf{A}, W_{nijk}, \tilde{W}_{nijk}$)
2:     $\mathbf{X} = \text{EquiLinear}(\mathbf{A})$               ▷ $\mathbf{A}$ contains aggregated messages
3:     $\mathbf{Y} = \text{EquiLinear}(\mathbf{A})$
4:     $\mathbf{O}_n = \left[ \sum_{ijk} \left( W_{nijk} \left\langle \langle \mathbf{X}_n \rangle_j \langle \mathbf{Y}_n \rangle_k \right\rangle_i + \tilde{W}_{nijk} \left\langle \langle \mathbf{X}_n \rangle_j \vee \langle \mathbf{Y}_n \rangle_k \right\rangle_i \right) \right] + \mathbf{Y}_n$
5:     **return O**
6: **end procedure**
7: ▷ $n$ is a feature dimension, $W, \tilde{W}$ are learnable, $\langle \mathbf{X} \rangle_i$ is the projection onto the $i$-th grade of $\mathbf{X}$

---

**Algorithm 6** Backbone update

1: **procedure** BACKBONEUPDATE($\{\mathbf{s}_i\}, \{\mathbf{G}_i\}, \{\mathbf{T}_i^{\text{rel}}\}, \{\mathbf{T}_i\}$)
2:     $b_i, c_i, d_i, \vec{\mathbf{t}_i} = \text{MLP}(\text{concat}\left(\mathbf{s}_i, \mathbf{G}_i, \mathbf{T}_i^{\text{rel}}\right))$
3:     $\mathbf{R}_i = \text{EmbedRotor}(b_i, c_i, d_i)$
4:     $\mathbf{S}_i = \text{EmbedTranslator}(\tilde{\mathbf{t}}_i)$
5:     **return** $\mathbf{T}_i \mathbf{R}_i \mathbf{S}_i$
6: **end procedure**

---

Finally, in the backbone update step, we predict an invariant frame update, just like in IPA, which when concatenated with the original frame transforms equivariantly:

$$\mathbf{T}_i \mathbf{T}_{update} \rightarrow \mathbf{T}_{global} \mathbf{T}_i \mathbf{T}_{update}.$$

Permutation equivariance is also maintained by the GAFL architecture, since we use message passing on the fully connected graph. In the setting at hand, however, we break this permutation equivariance intentionally by introducing positional encodings for the nodes, as done in models that rely on original IPA such as RFdiffusion.

## A.3   Experiments

### A.3.1   Definition of novelty and diversity

For each sampled length, we compute the pairwise Template Modeling (TM) score between designable backbones as a measure for similarity between folds [70] and report *diversity* as the TM score averaged over all lengths,

$$Diversity = \frac{1}{N} \sum_{l=1}^{N} \frac{1}{n_l(n_l - 1)} \sum_{i=1}^{n_l} \sum_{\substack{j=1 \\ j \neq i}}^{n_l} \text{TM}(\mathbf{x}_i, \mathbf{x}_j)$$

with the total number $N$ of sampled lengths, the number of designable samples $n_l$ at a given length, and the $\mathbf{x}_i$ and $\mathbf{x}_j$ being the $i$-th and $j$-th samples, respectively.

The resulting *diversity* score reports on how similar are sampled backbones to each other. We compare the structures of designable samples to the natural proteins found in the PDB database using FoldSeek [58] in TMalign mode, and define *novelty* as the average of the highest TM score calculated over all designable samples:

$$Novelty = \frac{1}{n} \sum_{i=1}^{n} \max_j \text{TM}(\mathbf{x}_i, \mathbf{x}_j)$$

with $n$ the number of all designable samples and $\mathbf{x}_j$ being the sample from the PDB database with the highest TM score to the sample $\mathbf{x}_i$

Since diversity and novelty are defined as averaged similarity scores, small values are desirable for the generation of structurally distinct and *de novo* proteins.

Table A.2: Performance of different models when generating 10 backbones for each length in $\{60, 61, \ldots, 128\}$. For strand and helix content, we choose the PDB dataset filtered for lengths 60 to 128 as reference. FrameFlow denotes the model trained on the PDB published in [69].

| Method | Designability ($\uparrow$) | Diversity ($\downarrow$) | Novelty ($\downarrow$) | Helix Content | Strand Content |
|---|---|---|---|---|---|
| PDB (128) | - | - | - | 0.36 (0.00) | 0.22 (0.00) |
| SCOPe(128) | - | - | - | 0.33 (0.00) | 0.26 (0.00) |
| Genie-SCOPe | 0.72 (0.02) | 0.38 (0.00) | **0.67** (0.00) | 0.66 (0.01) | 0.07 (0.01) |
| Genie-SwissProt | 0.79 (0.02) | **0.37** (0.00) | 0.68 (0.00) | 0.66 (0.01) | 0.09 (0.01) |
| FoldFlow-OT | **0.99** (0.00) | 0.49 (0.01) | 0.83 (0.01) | 0.87 (0.00) | 0.00 (0.00) |
| FrameDiff | 0.81 (0.02) | 0.44 (0.00) | 0.72 (0.00) | 0.39 (0.01) | 0.30 (0.01) |
| FrameFlow | 0.95 (0.01) | 0.38 (0.00) | 0.75 (0.00) | 0.53 (0.01) | 0.22 (0.01) |
| RFdiffusion | 0.98 (0.01) | 0.43 (0.01) | 0.8 (0.00) | 0.78 (0.00) | 0.07 (0.00) |
| GAFL | 0.96 (0.01) | 0.38 (0.00) | 0.78 (0.00) | 0.49 (0.01) | 0.27 (0.01) |

## A.3.2 Baselines used for comparison with GAFL

We generate backbones using RFdiffusion relying on publicly available weights with default settings (*noise_scale_ca*: 1) (GitHub RFdiff). For Genie, we use the model with the published weights trained either on the SCOPe or SwissProt datasets containing proteins of up to 128 or 256 residues long(GitHub Genie), and generate backbones with sampling for 1000 time steps, as it has demonstrated the best performance. FrameDiff was used with the newest published model weights (best_weights.pth) with sampling for 500 timesteps (GitHub FrameDiff). FrameFlow was used with the published weights trained on the same dataset as GAFL (GitHub FrameFlow). The originally model, denoted as FrameFlow* can be found at (GitHub FrameFlow (legacy)). We note that the GitHub repository supporting FrameFlow contains an implementation of minibatch OT [10], which we use for retraining FrameFlow. For FoldFlow we use the optimal transport model (foldflow-ot.pth) with inference annealing as suggested by [10] and sampling for 500 timesteps (GitHub FoldFlow).

## A.3.3 Results for small proteins

In order to evaluate GAFL's performance for small proteins in particular, we evaluate a range of models in the setting from the original FrameFlow [67] paper, where 10 backbones per length in $\{60, 61, \ldots, 128\}$ are generated. Apart form the models in Table 1, we also evaluate Genie [37], which was trained on backbones smaller than 300 residues and were therefore excluded from the evaluation in Table 1.

We find that GAFL and FrameFlow are the only models capable of generating highly designable backbones with diverse secondary structures for the considered length range, where GAFL outperforms FrameFlow in designability and secondary structure content. While FoldFlow and RFdiffusion, like GAFL, achieve designabilities over 90%, they generate backbones with 0% and 7% $\beta$-strand content, respectively, indicating a mode-collapse towards generating $\alpha$-helical structures. GAFL also outperforms both RFdiffusion and FoldFlow in terms of diversity and novelty. All other models considered have significantly lower designability. For Genie in particular, we observe good novelty but it has the lowest designability among the models considered and also under-represents beta strands.

## A.3.4 Results for long proteins

We also evaluate the performance of GAFL for generating proteins up to length 500, which enables to compare GAFL against vector field networks (VFN) [42], which at the time of writing have no published model weights. We thus evaluate GAFL and other baselines trained on the PDB with the same inference settings as in [42], sampling 5 protein backbones for each length in $\{100, 105, \ldots, 500\}$. We report the results in Table A.3. For VFN, we can only report on designability since [42] have different definitions for diversity and novelty respectively and do not calculate the secondary structure content of generated structures.

Table A.3: Comparison of GAFL with VFN, FrameDiff, and RFdiffusion for the generation of 5 protein backbones at each length $\in \{100, 105, \ldots, 500\}$. Values for VFN from [42].

| Method | Designability ($\uparrow$) | Diversity ($\downarrow$) | Novelty ($\downarrow$) | Helix Content | Strand Content |
|---|---|---|---|---|---|
| FrameDiff | 0.28 (0.02) | 0.44 (0.01) | 0.70 (0.01) | 0.57 | 0.17 |
| VFN | 0.44 | - | - | - | - |
| RFdiffusion | 0.71 (0.01) | 0.36 (0.00) | **0.68** (0.00) | **0.52** | **0.28** |
| FrameFlow* | 0.64 (0.02) | **0.33** (0.00) | 0.70 (0.01) | 0.64 | 0.15 |
| GAFL (Ours) | **0.74** (0.01) | 0.35 (0.00) | 0.74 (0.00) | 0.68 | 0.14 |

*Published model weights

We find that GAFL generates the highest fraction of designable backbones, while also yielding the highest diversity. At the same time RFdiffusion has the best novelty and generates structures with a secondary structure content which is closest to the PDB. The trend that GAFL outperforms RFdiffusion with respect to secondary structure content on small proteins but performs worse on longer proteins is also already apparent from Figure 3.

### A.3.5 Ablation for CFA on the PDB

As described in Section 4.5, we retrained FrameFlow in the same setup as GAFL with default hyperparameters from the repository in A.3.2. For both FrameFlow and GAFL, we performed three training runs with the hyperparameters reported in Appendix A.3.8. For all runs we selected checkpoints using our checkpointing criterion, described in Section 4.1. In Table 2, we report the performance of the best checkpoints per run.

As extension of Figure 3, we directly compare the designability of GAFL and the published Frame-Flow model by length in Figure A.8. We find that GAFL has higher designability for lengths smaller than 200 and greater than 250. Since we only generate 200 backbones for each length, the margins of error of the individual designabilities overlap, except for length 300. The total designability of GAFL, averaged over all 1000 backbones, however, is significantly better than that of FrameFlow (Table 1).

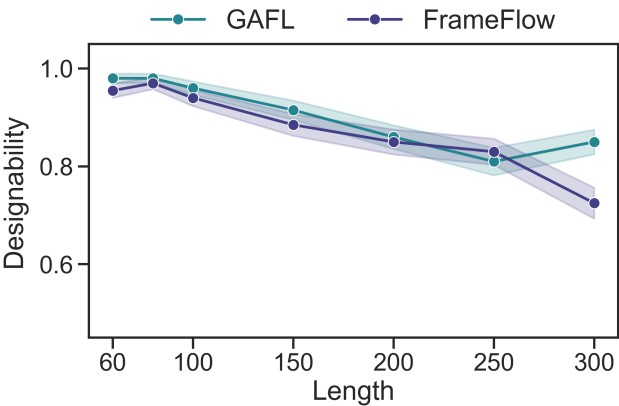

Figure A.8: Designability of GAFL and the published FrameFlow model as a function of length with standard errors obtained by bootstrapping the set of generated samples. 200 backbones are generated for each length, as in Figure 3.

### A.3.6 Timestep analysis of GAFL

We perform a timestep analysis, evaluating the best GAFL checkpoint with a different number of timesteps taken during inference, in order to judge its impact on the designability metric. To this end we generate 100 backbones for each length in $\{100, 150, \ldots, 300\}$, for different numbers of timesteps. We see that the performance beyond 200 timesteps stays almost constant, suggesting that 200 timesteps represents a good tradeoff between performance and inference speed.

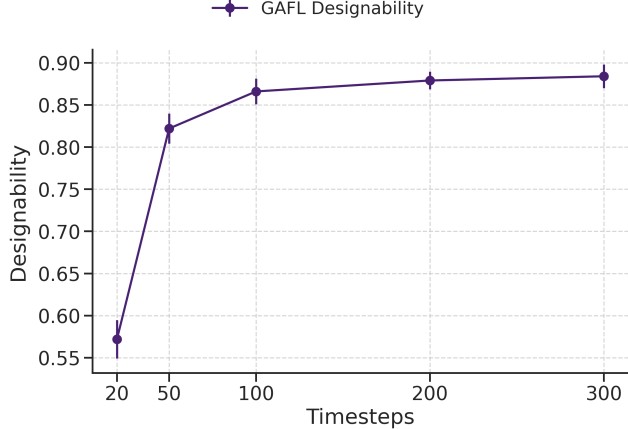

Figure A.9: Designability as a function of timesteps used during inference. 100 backbones for each length in $\{100, 150, \ldots, 300\}$ were sampled per each data point. Vertical lines denote the standard deviation.

### A.3.7 Inference Efficiency

We report the time needed for sampling protein backbones of length 100 on an NVIDIA A100 GPU with published model weights and default settings as described in A.3.2.

Table A.4: Time needed for sampling protein backbones of length 100 on an NVIDIA A100 GPU.

| Method | Timesteps | Time per Structure (s) |
|---|---|---|
| RFdiffusion | 50 | 21.0 |
| FrameDiff | 500 | 24.3 |
| FoldFlow | 500 | 24.2 |
| FrameFlow | 200 | 6.6 |
| GAFL | 200 | 8.8 |

### A.3.8 Training on PDB

On the PDB dataset, we train both GAFL and FrameFlow for 5200 epochs, where one epoch is defined as one iteration over all 4776 clusters, which we define as in FrameDiff by 30% sequence similarity. The learning rate is increased in 50 warmup steps to 0.0002 and then kept constant for 3500 epochs. From there we use a cosine-annealing schedule to decrease the learning rate to 0.0001 at epoch 5000. From epoch $N_{train} = 5000$ to $N_{train} + N_{select} = 5200$ we employ our checkpointing criterion described in Section 4.1 evaluating secondary structures and storing checkpoints every second epoch. We keep the $k = 30$ best checkpoints and filter them for checkpoints with a secondary structure content deviation of less than $d_{max} < 0.2$.

In order to select the best checkpoint we evaluate all filtered checkpoints by sampling 40 backbones for each length in $\{100, 150, 200, 250, 300\}$ and choosing the checkpoint with the highest designability. We then run new and independent inference runs for all experiments we conduct on the selected checkpoint.

### A.3.9 Training on SCOPe (Ablation)

On SCOPe, we train both GAFL and FrameFlow for 6500 epochs. We use a constant learning rate of 0.0001 for the whole procedure. From epoch $N_{train} = 4000$ to $N_{train} + N_{select} = 6500$ we use our ceckpointing criterion, storing checkpoints every 25 epochs. We keep the $k = 10$ best checkpoints also filtering them for checkpoints with a secondary structure content deviation of less than $d_{max} < 0.2$. We evaluate each filtered checkpoint by sampling 10 backbones for each length in $\{60, 61, \ldots, 128\}$ and visualize the resulting designability distribution in Figure A.10.

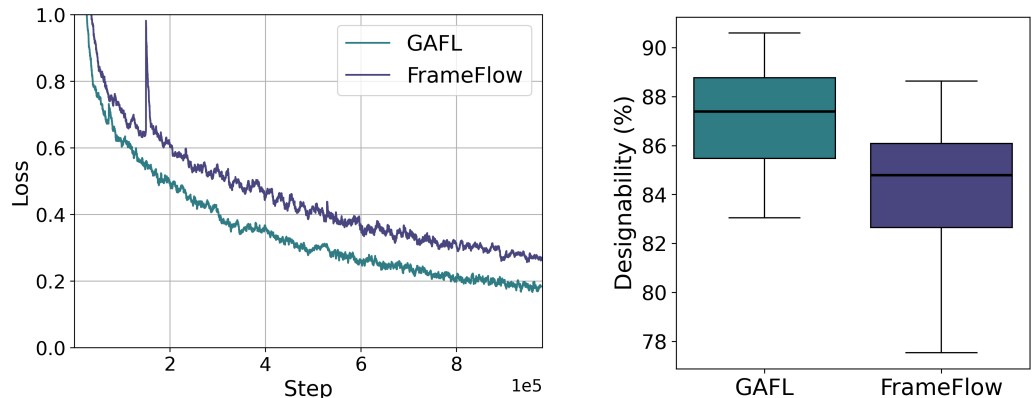

Figure A.10: Left: Total train loss averaged over flow matching times 0.5 to 0.75 on SCOPe for GAFL and FrameFlow. Right: Designabilities of 30 checkpoints sampled during three training runs on SCOPe for GAFL and retrained FrameFlow, respectively and filtered for a secondary structure content deviation of less than $d_{max} < 0.2$. For each checkpoint, we sample 10 backbones per length in $\{60, 61, \dots, 128\}$.

### A.3.10 Model hyperparameters

We report the most important hyperparameters of the CFA/IPA modules used in GAFL and FrameFlow respectively in Table A.5. For an extensive list of hyperparameters we refer to the respective config files on GitHub.

| Parameter | GAFL | FrameFlow |
|---|---|---|
| Node embedding size | 240 | 256 |
| Edge embedding size | 120 | 128 |
| Number of attention heads | 8 | 8 |
| Number of query/key points | 8 | 8 |
| Number of geometric value channels | 8 | 12 |
| Number of IPA/CFA blocks | 6 | 6 |

Table A.5: Comparison of the most important hyperparameters used for the CFA/IPA modules in GAFL and FrameFlow

The model hyperparameters of GAFL are choosen such that its total number of trainable parameters roughly equals that of FrameFlow as detailed in Table A.6.

| Model component | GAFL | FrameFlow |
|---|---|---|
| Embedding | 135 K | 150 K |
| CFA/IPA | 9.2 M | 8.4 M |
| Seq transformer | 4.5 M | 5.1 M |
| Node update | 1.0 M | 1.2 M |
| Edge update | 1.7 M | 1.9 M |
| BB update | 169 K | 10 K |
| Total | 16.7 M | 16.7 M |

Table A.6: Comparison between the number of parameters of GAFL and FrameFlow, broken down into contributions from different components.

### A.3.11 Memory consumption

Training on SCOPe with the model hyperparameters listed in A.3.10 and training hyperparameters given in A.3.9, results in a GPU memory consumption of 48.2 GB for FrameFlow and 59.5 GB for GAFL.

## A.4 Miscellaneous

### A.4.1 Societal Impact

Societal impact is considered mostly positive, as *de novo* protein design holds the promise to develop, for example, drugs against diseases, personalized therapies against cancer or also new nanomaterials, which out-weights potential risks, while of course security concerns remain, see e.g. Baker and Church [4].

### A.4.2 Implementation and Software Libraries

Our implementation is based on the implementation of FrameFlow [67, 69][4]. We will publish our code together with the camera ready version of this manuscript. The implementation is in the Python [59] programming language and uses the PyTorch framework [45] and further dependencies of FrameFlow: Numpy [30], Hydra [66], and SciPy [61].

---

[4]The implementation of FrameFlow is published under the MIT license at `https://github.com/microsoft/protein-frame-flow`

