# OpenReview forum: "Generating Highly Designable Proteins with Geometric Algebra Flow Matching"
_NeurIPS.cc/2024/Conference — NeurIPS 2024 poster_

### Official Review · Reviewer_zVgj · 2024-07-10

**Soundness:** 3
**Presentation:** 3
**Contribution:** 3
**Rating:** 6
**Confidence:** 4

**Summary:**

This paper tackles de-novo protein design, focusing on structure generation, where the goal is to generate novel protein backbones. The paper builds on top of FrameFlow, a well-established and popular method that uses flow matching, a residue frame-based protein backbone representation, and equivariant neural networks (invariant point attention and transformers). The proposed method, however, represents the oriented residue frames as elements of a projective geometric algebra. Different orientations and translations of the residue frames are essentially represented through reflections along a set of planes. The advantage of the approach is that it allows to use more complex operations on this algebra and design higher order messages in the message passing of the equivariant network. This can improve expressivity and performance, as the paper shows. The experiments are carried out on standard, relatively small scale, protein generation benchmarks, where the method demonstrates overall slightly favourable performance compared to baselines.

**Strengths:**

**Clarity:** While the math underlying the paper's main contribution is not easy, the high level intuitions are explained well and the paper is overall written clearly and presented well.

**Originality:** The paper essentially extends the Invariant Point Attention (IPA) layers used in previous works with ideas from geometric projective geometry. Such geometric projective geometry methods have been used before, but the integration into IPA and application for protein design seems to be new and original.

**Quality:** The paper is of overall good quality. The paper is well written, the experiments and baseline comparisons carried out as well as the ablations and analyses seem to be appropriate.

**Significance:** I think bringing novel ideas regarding new architectures and representations into the protein design field is timely. However, the results are only very incrementally better than the baselines and the scalability of these architectures is an open question (more below).

**Weaknesses:**

Methodologically, the paper seems interesting and novel, but the results are somewhat weak. On the important designability metric, the proposed geometric algebra flow matching method is only incrementally better than the most relevant FrameFlow baseline. This becomes clear especially in Table 5 in the Appendix, where the authors re-trained FrameFlow. On diversity and novelty, the method is not better than any baselines, sometimes actually slightly worse. Hence, the claimed performance boosts seem to be barely significant.

Moreover, the trained models are small and the generated protein backbones short. Whether the proposed approach is scalable remains an open question.

The authors said they kept the number of parameters of the model similar to the FrameFlow baseline, but more details here would be helpful, i.e. which components of the model have how many parameters. Furthermore, it would also be nice to report other properties of the proposed model. Does the proposed architecture converge quicker or slower? How does it affect memory consumption (which affects scalability)?

**Conclusions:** For these reasons, it is overall unclear how practically impactful the proposed method can be. These issues negatively affect the paper's significance. That said, despite these somewhat weak results, I think this is overall a well-written and interesting paper with insightful analyzes, and it proposes novel and original ideas. Hence, I am still leaning towards suggesting acceptance.

**Questions:**

The authors do not in detail write down their basic training hyperparameters, like learning rates, batch sizes, etc. I assume these are all taken exactly from FrameFlow? It would be great if the authors could clarify this.

**Limitations:**

Limitations and potential negative societal impact have been briefly, but appropriately discussed. I have no concerns.

---

> ### Author Rebuttal · Authors · 2024-08-07
>
> We thank the reviewer very much for their detailed and helpful review. Below we will discuss the comments line by line.
>
> **1. Scalability to larger proteins and datasets**
>
> As suggested by the reviewers, we trained GAFL on the PDB dataset of FrameDiff, which contains proteins of length up to 512 and is approximately ten times larger than SCOPe-128.
> We obtained very competitive results for sampling backbones of length up to 300, as is done in FrameDiff, FoldFlow and RF Diffusion.
> These new results demonstrate the scalability of GAFL to both large proteins and large datasets.
> For further details, please refer to Table A.1, Figures A.1 - A.3 and the part in the overarching comment on PDB training, where we argue that we consider GAFL a new state-of-the-art for protein backbone generation.
>
> **2. Performance on Diversity/Novelty**
>
> Regarding the performance of GAFL on diversity and novelty we would like to refer to the results achieved on the PDB dataset which we report in Table A.1.
> These results demonstrate that GAFL outperforms the other baselines on PDB with respect to diversity, and is on par with FrameDiff with respect to novelty.
>
> **3. Incrementally better results**
>
> While the designability improvement of GAFL over the published FrameFlow model is around ten percentage points (90.5\% vs 81\%), we indeed observed in our ablation that by using the original IPA architecture within our training setup (which includes the GAFL checkpoint selection algorithm), we can obtain a performance of up to 88\% designability for FrameFlow.
> We thus observe that the absolute improvement in percentage points due to the checkpoint selection algorithm is higher than that of the architecture.
> However, regarding that 88.2\% comprises already a high ratio of designable backbones, we consider an improvement from 88.2\% to 90.5\% a more significant step than an improvement by the same number of percentage points in lower regions of designablility.
> After all, the fraction of non-designable proteins was significantly reduced from 11.8\% to 9.5% by the GAFL model architecture.
>
> During the rebuttal period, we performed more extensive ablations to compare the architectures and to demonstrate that the difference in performance is significant.
> In Table A.3, we report the performance of three runs with different random seeds for different ablations of GAFL as median, min and max across these three runs.
> We observe that the GAFL architecture achieves consistently higher designabilities than the FrameFlow architecture: In this ablation study (Table A.3), row 2 corresponds to GAFL without data filtering, and rows 4 and 5 correspond to FrameFlow with and without the GAFL checkpoint selection criterion respectively.
>
> We also note that GAFL's good performance for longer proteins (see overarching comment, Table A.1, Figure A.1) sets it apart from other models trained on the PDB.
> To the best of our knowledge, similar results have not been achieved by any other architecture that does not rely on pretrained weights from folding models.
>
>
> **4. Number of parameters of GAFL**
>
> Although we applied fundamental changes to the model architecture by using multivector representations of the Geometric Algebra, we were able to retain a similar amount of parameters as the FrameFlow architecture, by reducing the node embedding size from 256 to 240 and the edge embedding size from 128 to 120.
>
> These changes result in the following number of parameters, which we will add to the appendix of the final version of the paper:
>
> || GAFL | FrameFlow |
> |--|--|--|
> |Embedding| 135 K | 150 K |
> |IPA| 9.2 M | 8.4 M |
> |Seq transformer| 4.5 M | 5.1 M |
> |Node update| 1.0 M | 1.2 M |
> |Edge update| 1.7 M | 1.9 M |
> |BB update| 169 K | 10 K |
> |Total| 16.7 M | 16.7 M |
>
> The complete hyperparameter config files of the model will be published along with the source code of the implementation with the final version of the paper.
>
> **5. Other properties of the model**
>
> It is challenging to judge the convergence properties of generative models in general since often the train loss has no absolute meaning and effects like mode collapse are difficult to notice from the training loss alone.
> This is also the case for GAFL.
> Still, we observe consistently better training loss values for GAFL in comparison to the FrameFlow architecture trained in our setup, which we provide as reference in Figure B.1.
>
> We observe that the training of GAFL is more stable in the sense that the distribution of designabilities for checkpoints selected with our checkpoint criterium is more narrow than for FrameFlow (see Figure B.2).
>
> Despite that both FrameFlow and GAFL have the same number of parameters, the need to store tensors of 16 dimensional multivectors increases the memory consumption of GAFL. More specifically, training on SCOPe with the same training hyperparameters results in the following GPU memory consumption:
>
> | Frameflow | GAFL    |
> |-----------|---------|
> | 48.2 GB   | 59.5 GB |
>
> **6. Training hyperparameters**
>
> We used the same hyperparameters as FrameFlow to enable a direct comparison between GAFL and FrameFlow and because we consider the hyperparameters as somewhat established. We will include the hyperparameters explicitly in a table in the supplementary in the camera-ready version of the manuscript.

---

> > ### Comment · Reviewer_zVgj · 2024-08-10
> > **Reply to Rebuttal**
> >
> > I would like to acknowledge that I have read the authors' rebuttal. I appreciate the extensive discussion as well as the additional results, in particular the experiments on the PDB. One more thing I would suggest, though, is to also re-train a standard FrameFlow model on the PDB data for the final version and include the results. FrameFlow is the most relevant and comparable baseline here and I believe FrameFlow will perform very well on the PDB, too. Although I would guess that GAFL would still be slightly better.
> >
> > I have raised my score and suggest acceptance of the work.

---

> > > ### Author Response · Authors · 2024-08-13
> > > **Author's response**
> > >
> > > We thank the reviewer for the suggestion of evaluating FrameFlow on the PDB. We will include this evaluation in the camera-ready version together with an ablation study on PDB similar to the one we performed on SCOPe.

---

### Official Review · Reviewer_yVxp · 2024-07-10

**Soundness:** 3
**Presentation:** 3
**Contribution:** 2
**Rating:** 6
**Confidence:** 4

**Summary:**

This paper introduces a new architecture based on Geometric Algebra Transformer (GAT) for protein backbone design. By using adapting the protein design IPA architecture with GAT, they train a SE(3) flow matching method to generate protein backbone. The contributions of the papers are essentially on the architecture. They are able to generate designable and diverse (in terms of secondary structure) proteins. They evaluated their proposed architecture on the SCOPe dataset and against standard baseline.

**Strengths:**

First of all, the paper is clearly written, clearly presented and easy to read. I really like the idea to revisit the IPA architecture with more recent transformer architectures that were developed for SE(3) data.

**Weaknesses:**

The weaknesses I see are ont the performed experiments and ablation study.

i) I think the authors should have run experiments on the PDB dataset because it makes the comparison between their method and other methods trained on the PDB datasets incomparable (Table 2). As some models were trained on longer proteins, their performance might not be optimal compared to GAFL.

ii) I would like to see the model trained on longer proteins. Currently, the largest protein size is 128 which is very small. Some of the competitors were trained on proteins of much larger size. For instance, FrameDiff and FoldFlow were trained on size up to 384. I wonder if the proposed architecture would still have been performing on longer proteins especially on the generated secondary structure. Maybe it would be harder for it and that would be a valuable input about the limits of the current method. If the authors retrain their method on the PDB dataset, I would encourage them to increase the protein length up to 384.

iii) I would like to see in the ablation study (Table 3) the use of different body order. The body order was set up at '3' and I would like to see the performance of when it is at least '2'.

iv) Some important papers should be discussed. Recently [Mao et al.,] introduced a new architecture to improve IPA with frame-diff. The comparison of GAFL with this method would strengthen the proposed manuscript.


[Mao et al] De novo Protein Design Using Geometric Vector Field Networks, ICLR 2024.

**Questions:**

See weakness section.

While the paper proposes a very interesting idea, I believe some additional experiments, comparisons and ablations are needed for it to pass the acceptance bar at NeurIPS. I am willing to improve my score if the reviewers are able to answer the limits I see in the current manuscript.

**Limitations:**

Yes.

---

> ### Author Rebuttal · Authors · 2024-08-07
>
> We thank the reviewer for their time invested in reading the paper and for their informative and constructive feedback. Below we will discuss the comments line by line.
>
> **1. Training on longer proteins**
>
> During the rebuttal period, we trained GAFL on longer proteins (on the FrameDiff dataset of proteins up to length 512) and obtained very competitive results for sampling backbones of length up to 300, as is done in FrameDiff, FoldFlow and RFdiffusion.
> As we argue in the overarching comment, with the new results, we consider GAFL as state-of-the-art for protein backbone generation of various lengths trained solely on the PDB or SCOPe-128 dataset.
> For further details, we refer to Table A.1, Figures A.1 - A.3 and the part in the overarching comment on PDB training.
>
> **2. Incomparability to methods trained on the PDB datasets**
>
> Having trained GAFL on the PDB dataset from FrameDiff makes it more comparable with FrameDiff, FoldFlow and RFdiffusion, which have been trained on (largely) the same dataset.
> In Table A.2, we report the values of both GAFL trained on the PDB and on SCOPe-128 using the same lengths and metrics as in Table 2 of the original submission.
> GAFL trained on the PDB with longer proteins outperforms GAFL trained on the SCOPe-128 dataset on designability, diversity and novelty and has competitive secondary structure distribution.
> Thus we find that, for GAFL, training on larger proteins has no harming effect on the performance for smaller proteins.
>
> **3. Ablation study for different body order**
>
> The second line in the ablation study in the original submission (Table 3) already corresponds to a body order of 2:
>
> Messages after the application of the *GeometricBilinear* layer (see equation 12) are of body order 2.
> Higher body order messages are implicitly constructed by repeatedly taking products between such 2 body messages as explained in equation 14, which we implemented in GAFL for body order 3 through algorithm 5.
> In the second line of the ablation study in Table 3 without higher order messages, we essentially remove this layer from the network, resulting in a message passing step which uses body order 2.
> We will clarify this in the final version of the paper and thank the reviewer for the remark.
>
>
> **4. Comparison with vector field networks**
>
> We thank the reviewer for bringing the paper of [Mao et al. 2024] to our attention. This paper discusses interesting modifications to IPA, which are in some sense complementary to ours and in principle allow for a combined model that uses both methods as we discuss below.
>
> *Geometric feature representation*
>
> The vector field networks (VFN) in [Mao et al. 2024] use a set of virtual atoms as geometric features, i.e. a set of point features $\\{\\vec{q}_k \\in \\mathbb{R}^3\\}$. GAFL, on the other hand, uses multivector valued features $\\{\\mathbf{V}_k \\in \\mathbb{G}^{3,0,1}\\}$ in the projective Geometric Algebra (PGA). These contain points as subrepresentation but are also able to represent lines, planes and most importantly frames i.e. elements of the special Euclidean group SE(3).
>
> *Residue interactions*
>
> The central part of VFN is how it models interactions between residue frames. To this end VFN introduces the "vector field operator", which models residue interactions in a common local reference frame and thus also allows to utilize nonlinearities such as radial basis functions.
>
> GAFL uses the canonical bilinear operations of PGA namely the geometric product and the join to model interactions between residue frames. In contrast to VFN, which only considers 2-body interactions, GAFL also models higher body order interactions (specifically n=3 in the published version) similar to the construction in MACE [1].
>
> *Scope of modifications*
>
> The modifications of GAFL focus on learning more expressive geometric node features, which are able to parametrize the target space of residue frames. In the attention mechanism of IPA we thus only modify the calculation of attention values $\\mathbf{V}_i$ and their subsequent manipulations.
>
> In VFN additionally the calculation of attention scores $a_{ij}$ is modified, while GAFL uses the original attention mechanism of IPA.
>
> *Conclusion*
>
> Both models try to enhance the geometric expressivity of IPA, relying on different representations and interaction layers respectively. Both modifications could be built into one model by adding the improved calculation of attention scores via vector field operators to the GAFL architecture. We think that this would be an interesting line of research and look forward to exploring it in the future.
>
> We will add above discussion to the camera-ready version of the manuscript.
>
> **GAFL outperforms VFN for unconditional backbone generation**
>
> We also note that GAFL trained on the PDB outperforms VFN on the metric reported in [Mao et al. 2024].
> As done there, we sample 5 protein backbones for each length in [100,105,...,495,500] for 500 timesteps and re-fold 8 sequences predicted by ProteinMPNN.
> We obtain a designability score of 72 for GAFL compared to 56 reported in [2] (see overarching comment).
> Unfortunately, VFN's model weights are not published and a different metric for diversity and novelty is reported, thus we can only compare with the designability value reported in the paper.
>
> [1] Batatia I. et al., MACE: Higher Order Equivariant Message Passing Neural Networks for Fast and Accurate Force Fields (2022)
>
> [2] Mao et al., De novo protein design using geometric vector field networks (2024)

---

> > ### Comment · Reviewer_yVxp · 2024-08-09
> > **Thank you for the rebuttal**
> >
> > I have read the rebuttal. I thank the authors for their answer and I hope they will add the novel experiments and discussions to VFN to their manuscript. I have raised my score.

---

### Official Review · Reviewer_ZJDu · 2024-07-11

**Soundness:** 3
**Presentation:** 3
**Contribution:** 3
**Rating:** 7
**Confidence:** 5

**Summary:**

This paper extends frame-based protein backbone generation with projective geometric algebra. It allows higher-order geometric tensors in frame modeling as seen in EGNNs. Based on FrameFlow, it demonstrates great designability with relatively small increases in computational consumption.

**Strengths:**

- Although not the first work to introduce geometric algebra to proteins or molecules, it is the first to leverage its advantages in protein backbone generation, achieving competitive results.
- Introducing concepts from EGNNs like MACE is innovative and effective.
- The inclusion of DSSP distribution comparisons between different methods is informative.

**Weaknesses:**

- It is unclear whether this Geometric Algebra Extension maintains SE(3) invariance/equivariance and permutation invariance.
- In Tables 1 and 2,  what is the length stride used for generation?
- It would be interesting to see GAFL's performance when trained on the PDB dataset and its comparison with RFDiffusion.
- GAFL appears to be more complex than the original IPA. Information on its training consumption and runtime is missing. Additionally, it is important to know if GAFL is sensitive to the timesteps used for generation.

**Questions:**

It’s coherent to see GAFL has better designability and less diversity/novelty since EGNNs are well-suited to model the structure’s geometric nature. I would like to see how this architecture performs on protein function prediction or structure prediction.

**Limitations:**

Limitations are discussed in the paper.

---

> ### Author Rebuttal · Authors · 2024-08-07
>
> We thank the reviewer for their time invested in reading the paper and for their constructive and helpful review. Below we will discuss the comments line by line.
>
> **1. SE(3) and permutation equivariance of GAFL**
>
> As in the original IPA formulation, SE(3) equivariance of the GAFL architecture is based on expressing geometric features in canonically induced local frames, as described from lines 211 to 216 in the original submission.
>
> More specifically, going through the architecture step by step, the attention scores are calculated using the $L_2$ norm of the difference of point features, which is E(3) invariant.
> Equivariance of the message aggregation step is guaranteed by the expression $T_i^{-1} \circ T_j \circ \vec{v}_j^{hp}$ in line 11 of algorithm 1, where $\\vec{v}_j^{hp}$ are SE(3) invariant point values and the frames $\\{T_i\\}$ transform according to $T_i \\rightarrow T^{global} \\circ T_i$ such that the whole expression remains invariant:
>
> $T_i^{-1} \\circ {T^{global}}^{-1} \\circ T^{global}\\circ T_j \\circ \\vec{v}^{hp}_j = T_i^{-1} \\circ T_j \\circ \\vec{v}^{hp}_j.$
>
> In GAFL, we do not modify the calculation of attention scores from IPA, which means that their invariance remains ensured.
> During message passing, we use the same construction as above (see line 11 of algorithm 2), but use a different representation of SE(3), namely multivector features instead of point features.
> The choice of representation, however, does not influence the invariance of the whole expression.
> Also the relative frame transformations $\\mathbf{T}_i^{-1} \\mathbf{T}_j$  we compute are invariant. All subsequent layers including the *GeometricBilinear* layer and the *ManyBodyContraction* layer operate exclusively on invariant node features, hence overall equivariance is retained throughout those layers.
> Finally, in the backbone update step, we predict an invariant frame update, just like in IPA, which when concatenated with the original frame transforms equivariantly:
>
> $\\mathbf{T}_i\\mathbf{T}^{update} \\rightarrow \\mathbf{T}^{global}\\mathbf{T}_i\\mathbf{T}^{update}.$
>
> Permutation equivariance is also maintained by the GAFL architecture, since we use message passing on the fully connected graph.
> In the setting at hand, however, we break this permutation equivariance intentionally by introducing positional encodings for the nodes, as done in models that rely on original IPA such as RFdiffusion.
>
> We thank the reviewer for this suggestion and will add this clarification about equivariance to the camera-ready version of the paper.
>
> **2. Length stride in tables 1 and 2**
>
> For both tables, we sampled ten backbones for each length in [60,61,...,127,128].
> For the new results from training GAFL on the PDB, we report the results for sampling 200 backbones for each length in [100,150,200,250,300] (1,000 backbones in total), as is done in FrameDiff, FoldFlow and RFDiffusion in Table A.1.
>
> **3. Training on PDB**
>
> During the rebuttal period, we trained GAFL on the PDB dataset and obtained very competitive results, which we report in Table A.1 and Figures A.1 - A.3.
> As we argue in the overarching comment, due to the new results, we consider GAFL as state-of-the art for generating protein backbones trained solely on the PDB or SCOPe-128.
> For further details, please refer to the overarching comment.
>
> **4. Training consumption and runtime**
>
> The introduction of bilinear products which operate on the 16 dimensional algebra indeed leads to an increase in runtime of GAFL compared to FrameFlow. We already report inference time in Table 6 in the supplementary of the original submission, which is about 33\% higher than the one of FrameFlow but still approximately a factor of five below the other baselines if evaluated with 100 timesteps and a factor of three if evaluated with 200 timesteps as in Table A.1. For identical training hyperparameters, the training time per epoch of GAFL and FrameFlow on one A100 80 GB are:
>
> | Frameflow | GAFL   |
> |-----------|--------|
> | 22 min    | 34 min |
>
> Since Clifford networks are such a novel type of architecture, there are still no optimized GPU kernels available, as mentioned in [1]. Thus future development in this field might as well increase the runtime performance of GAFL.
>
> **5. Timestep analysis**
>
> During the rebuttal period, we conducted an analysis on the influence of the number of inference timesteps on designability of the sampled backbones for both the GAFL model trained on SCOPe-128 and for the GAFL model trained on PDB.
> In Figure A.4, the designability of five sampled backbones for each length in [60,61,...,127,128] is shown for a different number of inference timesteps.
> We observe that both models already have competitive designabilities for 50 timesteps and that the designability does not improve much when going beyond 200 timesteps.
> This speaks for the efficiency of our Flow Matching formulation, in which we aim to learn rectified flows that transform noise to data on approximately straight paths as explained in section 2.2 of the paper.
>
> For other metrics such as diversity and novelty, we did not observe a strong dependency on the amount of timesteps.
>
> **6. Application to protein function and structure prediction**
>
> We regard the main contribution of the present work to be the introduction of the novel architecture (and of a state-of-the-art model for protein backbone generation).
> In extensive ablations (Table A.3, original too), we have shown that using the GAFL architecture instead of original IPA is beneficial for protein backbone generation.
> As this task relies on capturing the geometry of protein backbones, we consider our work as demonstration of the potential of using GAFL for tasks beyond protein generation as well, such as protein folding or conformational ensemble prediction.
>
> [1] Ruhe D. et al. Clifford Group Equivariant Neural Networks (2023)

---

> > ### Comment · Reviewer_ZJDu · 2024-08-09
> >
> > Thanks for the rebuttal. Hope to see more comprehensive analysis on PDB in the future version. I've raised my score.

---

### Official Review · Reviewer_5KkW · 2024-07-11

**Soundness:** 3
**Presentation:** 3
**Contribution:** 2
**Rating:** 6
**Confidence:** 3

**Summary:**

This work focuses on the protein backbone generation task and improves FrameFlow, a flow-based protein backbone generation framework, based on geometric algebra. More specifically, this work proposes to represent residue frames with the elements of projective geometric algebra, which allows for the usage of higher-order message passing based on bilinear geometric products. The results in terms of designability, diversity, and the distribution of secondary structure distribution are all good compared with various baselines.

**Strengths:**

- The proposed model architecture is effective according to the experimental results. It is challenging to achieve both high designability and proper distribution of secondary structures, and this work has well addressed the problem.
- The presentation of the paper is good. The algorithms between the original version of Framediff and the proposed version show the changes clearly.

**Weaknesses:**

- The novelty of this work is limited. It seems that the application of geometric-algebra-based models on the protein backbone design task. The author might show the novelty of the architecture compared with the existing ones, though they may still not be used in this task.
- Lack of theoretical analysis or understanding of the proposed architecture. It would be more solid if the author could theoretically show the superiority of the proposed architecture compared with the original IPA.
- Lack of experiments. There are many other settings that are more useful in practical scenarios beyond unconditional backbone generation, such as motif scaffolding. The authors can refer to [1] for more details.
- It would be better if the author could provide a more extensive background of geometric algebra considering the audience of this paper may hold a background in machine learning for protein engineering. This would also make this paper more self-containing.


References:

[1] Huguet, G., Vuckovic, J., Fatras, K., Thibodeau-Laufer, E., Lemos, P., Islam, R., Liu, C.H., Rector-Brooks, J., Akhound-Sadegh, T., Bronstein, M. and Tong, A., 2024. Sequence-Augmented SE (3)-Flow Matching For Conditional Protein Backbone Generation. arXiv preprint arXiv:2405.20313.

**Questions:**

See weaknesses.

Besides, it seems GAFL is trained on the smaller dataset (SCOPe-128). I wonder its performance if it is trained on the same dataset (the larger one) used in FrameDiff.

**Limitations:**

The authors have discussed the limitations.

---

> ### Author Rebuttal · Authors · 2024-08-07
>
> We thank the reviewer for their constructive and helpful review. Below we will discuss the comments line by line.
>
> **1. Novelty of the architecture**
>
> We consider the proposed model architecture as novel from a theoretical point of view since, to the best of our knowledge, it comprises the first method that uses multivectors from Geometric Algebra as feature representation in a local frame framework.
> Besides that, it is also, to the best of our knowledge, the first time an architecture that uses Projective Geometric Algebra is applied in the context of protein design.
>
> *Differences to other architectures with Geometric Algebra*
>
> All previous architectures that relied on Geometric Algebra (e.g. [5,6]) have employed Geometric Algebra layers for their equivariance properties and were formulated as E(3) equivariant graph neural networks.
> In our case, equivariance is already ensured by working in the canonical local frames induced by the protein backbone, as in the original IPA. This enables to use more expressive layers that do not need to be E(3) equivariant.
> We use Geometric Algebra layers not for obtaining equivariance, but for the geometric inductive bias that multivector representation, join, and geometric product offer in contrast to e.g. MLPs on coordinate vectors.
>
> On top of that, we construct higher order messages that represent not only pairwise relationships but also the relationship of three or more nodes. This approach was introduced in MACE [4] but has never been applied together with Geometric Algebra or with local frames.
>
> *Differences to other local frame architectures*
>
> Local frames in principle enable the use of arbitrary SE(3) representations.
> Most often, vector-, or more generally tensor-representations are used, for instance in IPA and [3].
> The GAFL architecture, on the other hand, leverages the multivector representation of Geometric Algebra, which increases expressivity as we explain below and sets it apart from other local frame architectures.
> The performance increase we see over the original IPA formulation suggests that using a multivector representation might also be beneficial for other architectures that rely on local frames.
>
> **2. Theoretical analysis of the architecture**
>
> One can explicitly show that GAFL contains the original IPA based architecture as a special case, meaning that every function parametrized by IPA can also be parametrized by GAFL, which makes GAFL more expressive than the original IPA.
>
> To see this one can note that the original point representation of IPA is a subrepresentation of the multivector representation used in GAFL, where points can be embedded as trivectors (see e.g. chapters 2.5 and 6 of [2]). The parametrization of IPA is recovered by choosing weights such that the model only operates on this subrepresentation and ignoring any additionally added layers by setting them to the identity.
>
> The main theoretical motivations, which make this generalization of IPA actually favorable are:
>
> * The use of the geometric product and join as interaction between residues in the message passing step allows to capture more information on the geometric relation between them than the simple linear aggregation of point features which is done in IPA. As examples these bilinear products are able to compute quantities like distances, angles, areas, and volumes between geometric objects represented in the algebra (see [2]).
>
> * Since PGA enables the representation of frames, residue frames can be used as node/edge features as we do by calculating $\mathbf{T}_i^{-1}\mathbf{T}_j$.
>
> **3. More background on Geometric Algebra**
>
> We agree that more background on Geometric Algebra would help to better follow the theoretical reasoning behind the proposed architecture.
> We will thus add a self-contained introduction to Geometric Algebra to the appendix, following parts of the introductory presentations in [1,2], which we find well-suited for understanding the Geometric Algebra related theory part in more depth.
>
> **4. Lack of experiments / practical scenarios**
>
> We consider the task of unconditional backbone generation as well suited for benchmarking the proposed novel architecture, and for demonstrating the benefits of incorporating the proposed Geometric Algebra layers for representing the geometry of protein backbones.
> While we consider this as the main contribution of the paper (see general remarks), we believe that the GAFL architecture can also be integrated into motif scaffolding or conformational ensemble prediction as these tasks also rely on capturing the backbone geometry.
>
> Further we want to note that [1] was published after the paper submission deadline, however, GAFL can also be incorporated in the method presented there, which uses original IPA.
>
> Following the suggestions of the reviewers, we performed additional experiments during the rebuttal period, including training on the PDB dataset, a more extensive ablation (Table A.3) and a number-of-timesteps analysis (Figure A.4).
>
> **5. Training on the dataset used in FrameDiff**
>
> We trained GAFL on the FrameDiff dataset during the rebuttal period and obtained very competitive results for sampling backbones with lengths of up to 300, as is done in FrameDiff and FoldFlow.
> Due to the new results, we consider GAFL as state-of-the-art for protein backbone generation trained solely on the PDB or the SCOPe-128 dataset.
> For further details, we refer to Table A.1, Figures A.1 - A.3.
>
> **References:**
>
> [1] Doran, C., Lasenby, A., Geometric Algebra for physicists (2003)
>
> [2] Dorst, L., A Guided Tour to the Plane-Based Geometric Algebra PGA Leo Dorst (2020)
>
> [3] Weitao Du et al., Se(3)equivariant graph neural networks with complete local frames (2022).
>
> [4] Batatia I. et al., MACE: Higher Order Equivariant Message Passing Neural Networks for Fast and Accurate Force Fields (2022)
>
> [5] Ruhe D. et al., Clifford Group Equivariant Neural Networks (2023)
>
> [6] Brehmer J. et al., Geometric Algebra Transformer (2023)

---

> > ### Comment · Reviewer_5KkW · 2024-08-13
> >
> > Thanks for your response!
> >
> > The responses address most of my concerns.
> > I have raised my score.

---

### Author Rebuttal · Authors · 2024-08-07

We thank all reviewers cordially for reviewing our paper and appreciate their helpful comments.
We are happy to read that the reviewers find our approach novel (ZJDu,zVgj) and effective (ZJDu,5KkW), and consider the topic as important (zVgj,yVxp) and challenging (5KkW) and the paper to be clearly written and well presented (zVgj,yVxp,5KkW).
Following the constructive remarks of all reviewers, we conducted several new experiments during the rebuttal period and obtained new results, which, we think, greatly improve the impact of our submission.

We retrained GAFL on the PDB dataset, which is roughly ten times larger than the SCOPe-128 dataset considered in the original submission and comprises proteins with lengths of up to 512.
For all protein lengths considered, GAFL outperforms the other baselines that do not rely on pretrained folding models on the task of unconditional generation of highly designable and structurally diverse backbones.

In the following, we will elaborate on that and address other points of general relevance.

**General remarks on the scope of the paper**

We consider our main contributions to be the introduction of

a) an enhanced architecture for representing protein structure

b) a checkpoint selection algorithm that prevents the mode collapse towards alpha helices of generative models with very high designability and

c) a new state-of-the-art model for generation of highly designable and yet structurally diverse protein backbones that
    is trained on the PDB or SCOPe-128.

**Training on the PDB dataset**

Several reviewers suggested to train GAFL on the Protein Data Bank (PDB) dataset, which comprises longer proteins and is approximately ten times bigger than the SCOPe-128 dataset used in the original submission.
We used the same PDB dataset used for training FrameDiff, which is composed of proteins with lengths of up to 512 clustered by 30\% sequence similarity.
In the following, we report preliminary results from training GAFL on this dataset for three days on two A100 GPUs during the rebuttal period with the same hyperparameters as in the original submission.

As in FoldFlow and FrameDiff, we evaluate GAFL for proteins of length up to 300 on the metrics stated in the original submission and report the results in Table A.1.
GAFL outperforms (or is on par with) all baselines with respect to diversity, novelty, helix and strand content.
On designability, GAFL is only surpassed by FoldFlow-OT, which performs poorly on all other metrics considered (e.g. it samples backbones with strand content below 0.5\%) and by RFdiffusion, which relies on weights from the folding model RoseTTaFold (trained for 1792 V100-GPU days [1]) and is trained for 21 A100-GPU days [2] on a dataset that is significantly larger than the PDB [2, SI 1.4].
Moreover, since GAFL uses only 200 timesteps for the backbones sampled in Table A.1, its inference time is the lowest among all models considered; we also implemented a batched inference mode, which reduces the inference time to 3.1 s per sampled backbone of length 100 with 200 timesteps on an A100 GPU.
From the above we conclude that GAFL is a *current state-of-the* art model for unconditional sampling of highly designable protein backbones with robust secondary structure distribution among models trained solely on the PDB or the SCOPe-128 dataset.

We also investigate the performance by protein length and note that the mode collapse of RFdiffusion towards predominantly helical structures (that we already observed in our experiments in the original submission) persists for proteins below length 150, which is not the case for GAFL trained on the PDB.
Since in most protein design campaigns the goal is to incorporate the desired functionality into the smallest protein possible (the cost of synthetizing and testing proteins experimentally increases with their size), we consider this advantage of GAFL over RFdiffusion to be *highly impactful for future developments and real-world applications* of generative models for protein backbones.

Finally, we evaluate GAFL's performance also for proteins of length up to 500.
In the table below, we report designability scores of backbones of lengths [100,105,...,500] in percent for different numbers of timesteps and ProteinMPNN-sequences and compare with the recent model VFN [4] and the values reported for FrameDiff in [4].

|Designabilities averaged over lengths [100,105,...,500]||||
|-|-|-|-|
|Num Timesteps |500| 500|200|
|Num pMPNN Seq.| 100  |8| 100|
|*FrameDiff*| 38 | 28 | - |
|*VFN*| 56| 44 | - |
|*GAFL*|**72**|**53**|62|

GAFL outperforms both baselines with less timesteps required, while still achieving diversity of 0.35 and strand and helix content of 0.18 and 0.59 for all timesteps and number of sequences considered.
We do not report metrics apart from designability for the baselines as the model weights are not published and different metrics are reported in [4].

**Outlook**

We demonstrate that incorporating the proposed Geometric Algebra layers is beneficial for the task of unconditional protein backbone generation, for which representing the protein backbone geometry is crucial.
We consider this task as a fundamental benchmarking task for further development of application-related conditional generative methods like motif scaffolding.

Further, since the proposed method extends the widely used original IPA architecture, it may be integrated easily into other state of the art architectures and methods for general protein backbone related tasks such as folding or conformational ensemble prediction [1,3,4].

[1] Baek, M. et al. Accurate prediction of protein structures and interactions using a three-track neural network (2021)

[2] Watson, L. et al. De novo design of protein structure and function with RFdiffusion (2023)

[3] Jing, B. et al. AlphaFold Meets Flow Matching for Generating Protein Ensembles (2024)

[4] Mao et al., De novo protein design using geometric vector field networks (2024)

---

### Author Response · Authors · 2024-08-07
**Clarification on FoldFlow values in Table 2 of the original submission**

Note: The values reported in Table 2 in our original submission contained values for FoldFlow that were obtained without inference annealing.
If evaluated with inference annealing, the backbones sampled by FoldFlow-OT are highly designable (99.7\%) but the model underperforms on all other metrics: The strand content is below 0.5\%, which we interpret as mode collapse towards sampling mostly helical (86\%) structures, which is also reflected in poor diversity (0.49) and novelty (0.82) scores.

We will include these values in the final submission.

---

### Decision · Program_Chairs · 2024-09-25

**Decision:**

Accept (poster)

**Comment:**

This paper introduces a novel architecture called the Geometric Algebra Transformer (GAT) for protein backbone design. This work leverages geometric algebra in an interesting way. While previous works have explored geometric algebra in the context of proteins or molecules, this work is the first to focus on protein backbone generation, achieving results that are competitive with state-of-the-art methods. The GAT architecture achieves equivariance by combining higher-order message passing with the expression of geometric features in canonically induced local frames. This method allows for more expressive layers that do not require E(3) equivariance, as the geometric algebra layers provide inductive biases through multivector representation, join, and geometric product, which is advantageous w.r.t. traditional MLPs on coordinate vectors, inspired by the original IPA method.

The paper is well-written, with comprehensive experiments, baseline comparisons, ablations, and analyses that effectively support the proposed approach. The rebuttal answers were convincing. I would ask the authors to include the new experiments in the final version of the paper. I found the work a compelling case for considering using geometric algebra beyond E(3) equivariance (which is guaranteed by the canonical frame).